# On the Provable Generalization of Recurrent Neural Networks

**Lifu Wang**
Beijing Jiaotong University
Lifu_Wang@bjtu.edu.cn

**Bo Shen**[*]
Beijing Jiaotong University
bshen@bjtu.edu.cn

**Bo Hu**
Beijing Jiaotong University
hubo2018@bjtu.edu.cn

**Xing Cao**
Beijing Jiaotong University
caoxing@bjtu.edu.cn

## Abstract

Recurrent Neural Network (RNN) is a fundamental structure in deep learning. Recently, some works study the training process of over-parameterized neural networks, and show that over-parameterized networks can learn functions in some notable concept classes with a provable generalization error bound. In this paper, we analyze the training and generalization for RNNs with random initialization, and provide the following improvements over recent works:

(1) For a RNN with input sequence $x = (X_1, X_2, ..., X_L)$, previous works study to learn functions that are summation of $f(\beta_l^T X_l)$ and require normalized conditions that $||X_l|| \leq \epsilon$ with some very small $\epsilon$ depending on the complexity of $f$. In this paper, using detailed analysis about the neural tangent kernel matrix, we prove a generalization error bound to learn such functions without normalized conditions and show that some notable concept classes are learnable with the numbers of iterations and samples scaling almost-polynomially in the input length $L$.

(2) Moreover, we prove a novel result to learn N-variables functions of input sequence with the form $f(\beta^T[X_{l_1}, ..., X_{l_N}])$, which do not belong to the "additive" concept class, i,e., the summation of function $f(X_l)$. And we show that when either $N$ or $l_0 = \max(l_1, .., l_N) - \min(l_1, .., l_N)$ is small, $f(\beta^T[X_{l_1}, ..., X_{l_N}])$ will be learnable with the number iterations and samples scaling almost-polynomially in the input length $L$.

## 1 Introduction

In Deep Learning, the recurrent neural network (RNN) is well-known as one of the most popular models to model sequential data and is widely used in practice for tasks in natural language processing (NLP). One of the characters of RNN is that it performs the same operation for all the input of the sequence.

Consider a input sequence $x = (X_1, X_2, ..., X_L)$. A RNN with the form

$$h_l(x) = \phi(\boldsymbol{W} h_{l-1} + \boldsymbol{A} X_l), \tag{1}$$

---

[*]Corresponding Author.

35th Conference on Neural Information Processing Systems (NeurIPS 2021).

is trying to learn functions $f_l(X_1, X_2, ...X_l)$ as

$$
\begin{aligned}
h_1(x) &= f_1(X_1) \\
h_2(x) &= f_2(X_1, X_2) \\
&\vdots \\
h_L(x) &= f_L(X_1, X_2, ...X_L)
\end{aligned}
\tag{2}
$$

Due to the complex nonlinearity, the loss is generally non-convex, and it is very difficult to give a theoretical guarantee. Recently, there are some works Allen-Zhu et al. (2019b); Cao and Gu (2019); Allen-Zhu et al. (2019a); Du et al. (2019); Arora et al. (2019); Allen-Zhu et al. (2019c) trying to give a theoretical explanation that why gradient descent can allow an overparametrized network to attain arbitrarily low training error and ample generalization ability. These papers show that, under some assumptions, we have:

- *Multi-layer feed-forward networks Allen-Zhu et al. (2019b); Du et al. (2019) and recurrent neural networks Allen-Zhu et al. (2019c) with large hidden size can attain zero training error, regardless of whether the data is properly labeled or randomly labeled.*

- *For multi-layer feed-forward networks, functions with the form $F^*(x) = \sum_{r=1}^{C} \phi_r(\beta_r^T X), X \in \mathbb{R}^d, \beta_r \in \mathbb{R}^d, ||\beta_r|| = 1$ are learnable i.e. fitting the training data with a provably small generalization error, if $\phi$ is analytic and the "complexity" is low enough Allen-Zhu et al. (2019a); Arora et al. (2019); Cao and Gu (2019).*

- *The "complexity" of function $\phi$ can be measured by a matrix derived from the NTK (Neural Tangent Kernel) of the network Arora et al. (2019); Cao and Gu (2019).*

- *For recurrent neural networks Allen-Zhu and Li (2019a), if the input sequence is normalized, i.e., $x = (X_1, X_2, ..., X_L)$, $||X_1|| = 1$, $||X_l|| = \epsilon$ with $\epsilon$ very small, functions with the form $F^*(x) = \sum_{l=1}^{L} \sum_{r=1}^{C_l} \phi_{l,r}(\beta_{l,r}^T X_l)$ are learnable, where $m$ is the size of matrix $\mathbf{W}$, and $\mathscr{C} = \sum_{i=0}^{\infty} a_i R^i$ is a series representing the complexity of learnable functions.*

These works show the provable learning ability of deep learning. But there are still some important issues that were not addressed.

- Firstly, for RNNs, the method in Allen-Zhu and Li (2019a) requires a normalized condition for $\mathbf{A}$ and $X_l$ in (1) that $||\mathbf{A}X_l|| \leq \epsilon_x$ for all $l \leq L$ and shows that for a function $F^*(x)$ with the complexity $\mathscr{C}$, it is learnable with error $O(\epsilon_x^{1/3} \mathscr{C})$. Thus $||X_l||$ (or equally, $||\mathbf{A}||$) should be very small and the scale is dependent on the complexity of functions. The dependence of $||\mathbf{A}X_l||$ on $\mathscr{C}$ makes the results unrealistic in practice since generally the norm of input will not be so small.

- Secondly, the result in Allen-Zhu and Li (2019a) shows that RNNs can learn functions which are the summation of functions like $\psi(\beta_l^T X_l)$. But this is only a linear combination of the functions of the input at different positions and does not consider the nonlinear interaction of the inputs. One may ask, since $h_L(x)$ is a function of $\{X_1, X_2, ...X_L\}$, is it possible to go beyond and learn more complex functions?

In order to study these problems, we consider the binary classification problem: for every input $x_i$, the label ($+1$ or $-1$) of $x_i$ can be expressed by the sign of a target function $F^*(x_i)$. We consider Elman recurrent neural networks with ReLU activation

$$
\begin{aligned}
h_l(x) &= \phi(\mathbf{W}h_{l-1} + \mathbf{A}X_l) \\
f(\mathbf{W}, x) &= \mathbf{B}^T h_L(x) \in \mathbb{R}. \\
x &= (X_1, X_2, ..., X_L), X_l \in \mathbb{R}^d, \mathbf{W} \in \mathbb{R}^{m \times m}, \\
\mathbf{A} &\in \mathbb{R}^{m \times d}, \mathbf{B} \in \mathbb{R}^m, \phi(x) = \max(x, 0)
\end{aligned}
\tag{3}
$$

to learn two types of target functions:

- Additive Concept Class:

$$F^*(x) = \sum_{l=1}^{L} \sum_{r=1} \psi_{l,r}(\beta_{l,r}^T X_l / ||X_l||),$$

$$\psi_{l,r}(x) = \sum_{i=0}^{\infty} c_i x^i, \tag{4}$$

- N-variables Concept Class:

$$F^*(x) = \sum_{r} \psi_r(\langle \beta_r, [X_{l_1}, ..., X_{l_N}] \rangle),$$

$$\psi_r(x) = \sum_{i=0}^{\infty} c_i x^i. \tag{5}$$

For these two types of function, we study the following questions:

- Can RNN learn additive concept class functions (4) without the normalized condition with **reasonable complexity on the sequence size** $L$?

- Can RNN learn functions in N-variables Concept Class (5) which can not be written as the summation of $f(X_l)$ with **reasonable complexity on** $N$ **and** $L$?

**Our Result.** We answer the two questions and give a provable generalization error bound. Our results are stated as follows:

**Theorem 1** *(Informal) For a function $F^*(X_1, X_2, ..., X_L)$ with the form as in (4) or (5), there is a power series named the complexity $\mathscr{C}(F^*)$ dependent on the Taylor expansion coefficient in (4) and (5). For (4), $\mathscr{C}(F^*)$ is almost-polynomial in L. For (5), when N or $l_0 = \max(l_1, .., l_N) - \min(l_1, .., l_N)$ is small, $\mathscr{C}(F^*)$ is almost-polynomial in L. Under this definition of complexity $\mathscr{C}(F^*)$, $F^*$ is learnable using RNN with m hidden nodes and ReLU activation in (3) in $\mathcal{O}(\mathscr{C}(F^*)^2)$ steps with $\mathcal{O}(\mathscr{C}(F^*)^2)$ samples if $m \geq poly(L, \mathscr{C}(F^*))$.*

**Contribution.** We summarize the contributions as follows:

- *In this paper, we prove that RNN without normalized condition can efficiently learn some notable concept classes with **both time and sample complexity scaling almost polynomially in the input length** $L$.*

- *Our results go beyond the "additive" concept class. We prove a novel result that RNN can learn more complex function of the input such as N-variables concept class functions. And "long range correlation functions" with small N (e.g. $N = 2$, $f(\beta^T[X_l, X_{l+l_0}])$) are learnable with **complexity scaling almost polynomially in the input length** $L$ **and correlation distance** $l_0$.*

- *Technically, we study the "backward correlation" of RNN network. In RNN case, using a crucial observation on the degeneracy of deep network, we show that the **"backward correlation"** $\frac{1}{m}\langle \mathbf{Back}_l(x_i), \mathbf{Back}_l(x_j) \rangle$ **will decay polynomially rather than exponentially in input length** $L$. This shows the complexity of learning RNN with ReLU activation function is polynomial in the size of input sequence L.*

**Notions.** For two matrices $\boldsymbol{A}, \boldsymbol{B} \in \mathbb{R}^{m \times n}$, we define $\langle \boldsymbol{A}, \boldsymbol{B} \rangle = \text{Tr}(A^T B)$. We define the asymptotic notations $\mathcal{O}(\cdot), \Omega(\cdot), poly(\cdot)$ as follows. $a_n, b_n$ are two sequences. $a_n = \mathcal{O}(b_n)$ if $\limsup_{n \to \infty} |a_n/b_n| < \infty$, $a_n = \Omega(b_n)$ if $\liminf_{n \to \infty} |a_n/b_n| > 0$, $a_n = poly(b_n)$ if there is $k \in \mathbb{N}$ that $a_n = O((b_n)^k)$. $\widetilde{\mathcal{O}}(\cdot), \widetilde{\Omega}(\cdot), \widetilde{poly}(\cdot)$ are notions which hide the logarithmic factors in $\mathcal{O}(\cdot), \Omega(\cdot), poly(\cdot)$. $|| \cdot ||$ and $|| \cdot ||_2$ denote the 2-norm of matrices. $|| \cdot ||_1$ denote the 1-norm. $|| \cdot ||_F$ is the Frobenius-norm. $|| \cdot ||_0$ is the number of non-zero entries.

For elements $A_{i,j}, B_{i,j}$ of symmetric matrix $\boldsymbol{A}, \boldsymbol{B}$. We abuse the notion $A_{i,j} \succeq B_{i,j}$ to denote $\boldsymbol{A} \succeq \boldsymbol{B}$, i.e. $\boldsymbol{A} - \boldsymbol{B}$ is a positive semidefinite matrix.

## 2 Preliminaries

### 2.1 Function Complexity

For a analytic function $\psi(z)$, we can write it as $\psi(z) = c_0 + \sum_{i=1}^{\infty} c_i z^i$. We define the following notion to measure the complexity to learn such functions.

$$\mathscr{C}(\psi, R) = 1 + \sum_{i=1}^{\infty} i \cdot |c_i| R^i. \tag{6}$$

$$\mathscr{C}_N(\psi, R) = 1 + \sum_{i=1}^{\infty} L^{1.5N} C_1^N \cdot \sqrt{C_{N,i}} \cdot (i/N)^N \cdot |c_i| R^i \tag{7}$$

where $C_1 > 100$ is an large absolute constant and $C_{N,i}$ is the largest combination number $\frac{i!}{n_1! n_2! ... n_N!}$ for $n_1, n_2 ... n_N > 0, n_1 + n_2 + ... n_N = i$,

**Example 2.1** *Arora et al. (2019) Consider $\psi(z) = arctan(z/2)$. Then*

$$\psi(z) = \sum_{i=1} \frac{(-1)^{i-1} 2^{1-2i}}{2i-1} z^{2i-1} \tag{8}$$

*In this case,*

$$\mathscr{C}(\psi, 1) = 1 + \sum_{i=1}^{\infty} i \cdot |c_i| \leq 1 + \sum_{i=1}^{\infty} 2^{1-2i} \leq \mathcal{O}(1).$$

**Example 2.2** *In the case $N = 2$, $C_{2,i} = i$, $(i/2)^2 \leq i^2$. $\psi(z) = exp(z)$*

$$\mathscr{C}_2(\psi, 1) \leq 1 + \sum_{i=1}^{\infty} L^3 C_1^2 \pi i^{2.5}/i! \leq \mathcal{O}(1)$$

### 2.2 Concept Class

For the input sequence $\{X_l\}$, we assume $C_{min} \leq ||X_l|| \leq C_{max}$, for all $1 \leq l \leq L$ and $C_{max}/C_{min} \sim C_0$. Under this condition, we consider two types of target functions with the following form:
**Additive Concept Class.**

$$F^*(x) = \sum_{l=1}^{L} \sum_{r=1}^{C_l} \psi_{l,r}(\beta_{l,r}^T X_l / ||X_l||). \tag{9}$$

Here for all $l, r$, $\psi_{l,r}$ is analytic and $||\beta_{l,r}||_2 \leq 1$.

We define

$$\mathscr{C}(F^*) = L^{3.5} \sum_{l=1}^{L} \sum_{r=1}^{C_l} \mathscr{C}(\psi_{l,r}, C_0 \sqrt{L}), \tag{10}$$

to be the complexity of the target function.

**Remark 2.1** *If we consider function $\psi(\beta^T X_l)$ and $||X_l|| = 1$ for all l, the above complexity will become $\mathscr{C}(\psi, \mathcal{O}(\sqrt{L}))$. This is similar with that in Allen-Zhu and Li (2019a) but this complexity requirement is much weaker than that in Allen-Zhu and Li (2019a). For example, the complexity of $arctan(z/2)$ in Allen-Zhu and Li (2019a) is not finite, as shown in Arora et al. (2019).*

**N-variables Concept Class.**

$$F^*(x) = \sum_r \psi_r(\langle \beta_r, [X_{l_1}, ..., X_{l_N}] \rangle)/\sqrt{N} \max ||X_{l_n}||). \tag{11}$$

For all $r$, $\psi_{l,a,r}(x, y)$ is an analytic function $\psi_r(x) = c_0 + \sum_{i=1}^{\infty} c_i x^i$. $\beta_r \in \mathbb{R}^{dN}$, $||\beta_r||_2 \leq 1$. Let $l_0 = \max(l_1, .., l_N) - \min(l_1, .., l_N)$. We define

$$\mathscr{C}(F^*) = \min(L^2 \mathscr{C}_N(\psi_r, C_0 \sqrt{L}), L^{3.5} \mathscr{C}(\psi_r, 2^{l_0} C_0 \sqrt{L})). \tag{12}$$

**Remark 2.2** *The complexity $\sum_r \mathscr{C}_N(\psi_r, C_0\sqrt{L})$ and $\sum_r \mathscr{C}(\psi_r, 2^{l_0}C_0\sqrt{L})$ are exponential in $N$ and $l_0$ respectively. And $\mathscr{C}(F^*)$ is less or equal than both. Thus if either $l_0$ or $N$ is small, $\mathscr{C}(F^*)$ will be polynomial in L. Especially when $N$ is small(e.g. N=2), even if $l_0 = L - 1$, functions with the form $f(\beta^T[X_l, X_{l+l_0}])$ are still learnable with a low complexity.*

### 2.3 Results on Positive Definite Matrices and Functions

We say a function $\phi(\cdot, \cdot) : \mathbb{R}^d \times \mathbb{R}^d \to \mathbb{R}$ is positive definite if for all $n \in \mathbb{N}$, any $\{x_1, ..., x_n\} \subseteq \mathbb{R}^d, \{c_1, ..., c_n\} \subseteq \mathbb{R}$,

$$\sum_{i,j} c_i c_j \phi(x_i, x_j) \geq 0. \tag{13}$$

The following basic properties in chapter 3 of BergJens et al. (1984) are very useful in our proof.

**Proposition 2.1** *If $\phi(\cdot, \cdot)$ is positive definite function, let matrix $\boldsymbol{M} \in \mathbb{R}^{n \times n}$, $\{x_1, ..., x_n\} \subseteq \mathbb{R}^d$, and $M_{i,j} = \phi(x_i, x_j)$. Then $\boldsymbol{M}$ is a semi-positive definite matrix.*

**Proposition 2.2** *If $\phi_1(\cdot, \cdot)$ and $\phi_1(\cdot, \cdot)$ are positive definite, $\phi(x_i, x_j) = \phi_1(x_i, x_j) \cdot \phi_2(x_i, x_j)$ is also a positive definite function.*

**Proposition 2.3** *Let $\phi(\cdot, \cdot)$ be a positive definite function, and $\psi(x) = \sum_{i=0}^{\infty} c_i x^i$, $c_i \geq 0$. Then $\psi(\phi(\cdot, \cdot))$ is also a positive definite function.*

For a positive definite matrix $\boldsymbol{M} \in \mathbb{R}^{n \times n}$, there is a result in Arora et al. (2019),

**Proposition 2.4** *(Section E of Arora et al. (2019).) Let $\boldsymbol{X} = (x_1, ...x_n) \in \mathbb{R}^{d \times n}$ and $\boldsymbol{K}_p \in \mathbb{R}^{n \times n}$ is a matrix with $(K_p)_{i,j} = (x_i^T x_j)^p$. Suppose there is $\alpha > 0$, such that $\boldsymbol{M} \succeq \alpha^2 \boldsymbol{K}_p$. Let $y = ((\beta^T x_1)^p, ..., (\beta^T x_n)^p) \in \mathbb{R}^n$. We have $\sqrt{y^T (\boldsymbol{M})^{-1} y} \leq ||\beta||_2^p / \alpha$.*

## 3 Main Results

Assume there is an unknown data set $\mathcal{D} = \{x, y\}$. The inputs have the form $x = (X_1, X_2, ...X_L) \in (\mathbb{R}^d)^L$. $||X_l|| \leq \mathcal{O}(1)$ for all $1 \leq l \leq L$. For every input $x_i$, there is a label $y_i = \pm 1$.

The neural network with input $x$ is

$$\begin{aligned} h_0(x) &= \phi(\boldsymbol{M}_0), \\ h_l(x) &= \phi(\boldsymbol{W} h_{l-1} + \boldsymbol{A} X_l), \\ f(\boldsymbol{W}, x) &= \boldsymbol{B}^T h_L(x). \end{aligned} \tag{14}$$

Here $\boldsymbol{W} \in \mathbb{R}^{m \times m}$, $\boldsymbol{A} \in \mathbb{R}^{m \times d}$, $\boldsymbol{B}, \boldsymbol{M}_0 \in \mathbb{R}^m$. The entries of $\boldsymbol{M}_0$, $\boldsymbol{W}$ and $\boldsymbol{A}$ are respectively i.i.d. generated from $N(0, \frac{2}{m})$, $N(0, \frac{2}{m})$ and $N(0, \frac{2}{L^3 \cdot m})$. The entries of $\boldsymbol{B}$ are i.i.d. generated from $N(0, \frac{1}{m})$.

The goal of learning RNN is to minimize the population loss:

$$L_{\mathcal{D}}(\boldsymbol{W}) = \mathbb{E}_{(x,y) \sim \mathcal{D}} \ell(y \cdot f(\boldsymbol{W}, x)), \tag{15}$$

by optimizing the empirical loss

$$L_S(\boldsymbol{W}) = \frac{1}{n} \sum_{i=1}^{n} \ell(y_i \cdot f(\boldsymbol{W}, x_i)), \tag{16}$$

using SGD. Here $\ell(x) = \log(1 + exp(-x))$ is the cross-entropy loss. Consider the SGD algorithm on this RNN. Let the complexity $\mathscr{C}^*$ of $F^*(\cdot)$ be defined in (10) and (12). The 0-1 error for $\mathcal{D}$ is $L_{\mathcal{D}}^{0-1}(\boldsymbol{W}) = \mathbb{E}_{(x,y) \sim \mathcal{D}} \mathbb{1}\{y \cdot f(\boldsymbol{W}, x) < 0\}$. We have:

**Theorem 2** *Assume there is $\delta \in (0, e^{-1}]$. Supposing for $\mathcal{D} = \{x_i, y_i\}$, there is a function $F^*$ belonging to the concept class (9) or (11) such that $y_i \cdot F^*(x_i) \geq 1$ for all i. Let $\boldsymbol{W}^k$ be the output of*

---
**Algorithm 1:** Training RNN with SGD
---
**Input:** Data set $\mathcal{D}$, learning rate $\eta$.

The entries of $\boldsymbol{W}^0$, $\boldsymbol{A}$ are i.i.d. generated from $N(0, \frac{2}{m})$. The entries of $\boldsymbol{B}$ are i.i.d. generated from $N(0, \frac{1}{m})$.

**for** $t = 1, 2, 3 \ldots n$ **do**

    |    Randomly sample $(x_t, y_t)$ from the data set $\mathcal{D}$.

    |    $\boldsymbol{W}^t = \boldsymbol{W}^{t-1} - \eta \nabla_{\boldsymbol{W}^{t-1}} \ell(y_t \cdot f(\boldsymbol{W}^{t-1}, x_t))$.

**end**

---

*Algorithm 1. There is a parameter $m^*(n, \delta, L, \mathscr{C}^*) = poly(n, \delta^{-1}, L, \mathscr{C}^*)$ such that, with probability at least $1 - \delta$, if $m > m^*(n, \delta, L)$, there exits parameter $\eta = \mathcal{O}(1/m)$ that satisfies*

$$\frac{1}{n} \sum_{k=1}^{n} L_{\mathcal{D}}^{0-1}(\boldsymbol{W}^k) \leq \widetilde{\mathcal{O}}[\frac{(\mathscr{C}^*)^2}{n}] + \mathcal{O}(\frac{\log(1/\delta)}{n}). \tag{17}$$

**Remark 3.1** *This theorem induces that, to achieve* **population** $0 - 1$ **error***(rather than empirical loss) being less than $\epsilon$, it is enough to train the network using Algorithm 1 with $\widetilde{\Omega}((L \cdot \mathscr{C}^*)^2/\epsilon)$ steps. As defined in section 2.1 and 2.2, when N is small, for the two types of concept class, $(\mathscr{C}^*)^2$ is almost-polynomial in input length L. Thus they can be learned effectively.*

**Remark 3.2** *This theorem can also be generalized to "sequence labeling" loss such as $\frac{1}{n} \sum_{i=1}^{n} \sum_{l=1}^{L} \ell(y_i \cdot f_l(\boldsymbol{W}, x_i))$ with $f_l(\boldsymbol{W}, x) = \boldsymbol{B}^T h_l(x)$. This is because the matrix*

$$H_{i,j}^l = \frac{1}{m} \langle \nabla f_l(\boldsymbol{W}, x_i), \nabla f_l(\boldsymbol{W}, x_j) \rangle$$

*with different $l$ are almost "orthogonal" by a similar argument to (26) in Theorem 6. Then RNN can learn a function $f_l = sign(F_l^*(x))$ with $F_l^*(x)$ belonging to functions in section 2.2. See Remark G.1 in the supplementary materials.*

## 4 Sketch Proof of the Main Theorem

The first step to prove the main theorem 2 is the following generalization of Corollary 3.10 in Cao and Gu (2019).

**Theorem 3** *Under the condition of Theorem 2, let $n$ samples in the training set be $\{x_i, y_i\}_{i=1}^{n}$. $\widetilde{y} = [F^*(x_1), F^*(x_2), \ldots F^*(x_n)]^T$. Let $\boldsymbol{H}$ be a matrix with $H_{i,j} = \frac{1}{m} \langle \nabla_{\widetilde{W}} f(\widetilde{\boldsymbol{W}}, x_i), \nabla_{\widetilde{W}} f(\widetilde{\boldsymbol{W}}, x_j) \rangle$. The entries of $\widetilde{\boldsymbol{W}}$ are i.i.d. generated from $N(0, \frac{2}{m})$. If there is a matrix $\boldsymbol{H}^\infty \in \mathbb{R}^{n \times n}$ satisfying*

$$\boldsymbol{H} + \boldsymbol{\epsilon}^T \boldsymbol{\epsilon} \succeq \boldsymbol{H}^\infty \text{ with } ||\boldsymbol{\epsilon}||_F \leq 0.01/\mathcal{O}(\mathscr{C}^*), \tag{18}$$

*and $\sqrt{\widetilde{y}^T (\boldsymbol{H}^\infty)^{-1} \widetilde{y}} \leq \mathcal{O}(\mathscr{C}^*)$, there exits $m^*(n, \delta^{-1}, L, \mathscr{C}^*) = poly(n, \delta^{-1}, L, \mathscr{C}^*)$ such that, with probability at least $1 - \delta$, if $m > m^*$,*

$$\frac{1}{n} \sum_{k=1}^{n} L_{\mathcal{D}}^{0-1}(\boldsymbol{W}^k) \leq \widetilde{\mathcal{O}}[\frac{\widetilde{y}^T (\boldsymbol{H}^\infty)^{-1} \widetilde{y}}{n}] + \mathcal{O}(\frac{\log(1/\delta)}{n}). \tag{19}$$

**Remark 4.1** *In order to show Theorem 2 using this theorem, we need to carefully pick out the exponential parts of L. Using the methods in Allen-Zhu et al. (2019c) and Cao and Gu (2019), we can show that $m^*(L, n, \sqrt{\widetilde{y}^T (\boldsymbol{H}^\infty)^{-1} \widetilde{y}}) \geq poly(n, L, \sqrt{\widetilde{y}^T (\boldsymbol{H}^\infty)^{-1} \widetilde{y}})$ is enough. $\sqrt{\widetilde{y}^T (\boldsymbol{H}^\infty)^{-1} \widetilde{y}}$ is dealt with by calculating the forward and backward correlation in section 4.1.1 and 4.1.2.*

The proof of theorem 3 is in fact a combination of the results in Cao and Gu (2019) and Allen-Zhu et al. (2019c). The really matter thing is how large can $\sqrt{\widetilde{y}^T (\boldsymbol{H}^\infty)^{-1} \widetilde{y}}$ be. We can show that:

**Theorem 4** *Under the condition of Theorem 3, with probability at least $1 - \delta$, there exits matrix $\boldsymbol{H}^\infty$ satisfying (18) and*

$$\sqrt{\widetilde{y}^T (\boldsymbol{H}^\infty)^{-1} \widetilde{y}} \leq \mathcal{O}(\mathscr{C}^*). \tag{20}$$

Theorem 2 is a direct corollary of the above two theorems.

## 4.1 Calculation on Kernel Matrix

The proof of (20) relies on a direct calculation to construct a kernel matrix $\boldsymbol{H}^\infty$. We consider two input $x_i$ and $x_j$. Let $X_{i,l}$ and $X_{j,l}$ be the $l-th$ input of $x_i$ and $x_j$. Let $D_l \in \mathbb{R}^{m \times m}$ and $D'_l \in \mathbb{R}^{m \times m}$ be diagonal matrices that,

$$
\begin{aligned}
(D_l)_{k,k} &= \mathbb{1}\{\boldsymbol{W}h_{l-1}(x_i) + \boldsymbol{A}X_{i,l} > 0\} \\
(D'_l)_{k,k} &= \mathbb{1}\{\boldsymbol{W}h_{l-1}(x_j) + \boldsymbol{A}X_{j,l} > 0\}
\end{aligned}
\tag{21}
$$

$$
\text{Back}_l = BD_LW \cdots D_{l+1}W, \text{Back}'_l = BD'_LW \cdots D'_{l+1}W
\tag{22}
$$

Then

$$
\frac{1}{m}\langle \nabla_{\widetilde{W}} f(\widetilde{\boldsymbol{W}}, x_i), \nabla_{\widetilde{W}} f(\widetilde{\boldsymbol{W}}, x_j)\rangle = \frac{1}{m}\sum_{l,l'}\langle \text{Back}_l(x_i) \cdot D_l, \text{Back}_{l'}(x_j) \cdot D'_{l'}\rangle \cdot \langle h_l(x_i), h_{l'}(x_j)\rangle
\tag{23}
$$

Generally $H_{i,j} = \frac{1}{m}\langle \nabla_{\widetilde{W}} f(\widetilde{\boldsymbol{W}}, x_i), \nabla_{\widetilde{W}} f(\widetilde{\boldsymbol{W}}, x_j)\rangle$ is hard to deal with. However, in the $m \to \infty$ limit, we can use some techniques to do the calculation.

### 4.1.1 Forward Correlation

**Theorem 5** *For fixed $i, j$, under the condition in Theorem 3, with probability at least $1 - exp(-\Omega(\log^2 m))$,*

$$
|\langle h_l(x_i), h_l(x_j)\rangle - K^l_{i,j}| \le \mathcal{O}(l^{16} \cdot \log^2 m/\sqrt{m})
\tag{24}
$$

*And let $Q_l = \sqrt{(1 + \frac{1}{L^3}\sum_{k=1}^l ||X_{i,k}||^2) \cdot (1 + \frac{1}{L^3}\sum_{k=1}^l ||X_{j,k}||^2)}$,*

$$
\begin{aligned}
K^1_{i,j} &= Q_1 \cdot \sum_{r=0}^\infty \mu_r^2[(1 + \frac{1}{L^3}X_{i,1}^T X_{j,1})/Q_1]^r \\
K^l_{i,j} &= Q_l \cdot \sum_{r=0}^\infty \mu_r^2(\{\frac{1}{L^3}X_{i,l}^T X_{j,l} + K^{l-1}_{i,j}\}/Q_l)^r
\end{aligned}
\tag{25}
$$

*In the above equations, $\mu_r = \frac{1}{\sqrt{2\pi}}\int_0^\infty \sqrt{2}xh_r(x)e^{-\frac{x^2}{2}}dx$, $h_r(x) = \frac{1}{\sqrt{r!}}(-1)^r e^{\frac{x^2}{2}}\frac{d^r}{dx^r}e^{-\frac{x^2}{2}}$.*

### 4.1.2 Backward Correlation

**Theorem 6** *For $l \ne l'$, with probability at least $1 - exp(-\Omega(\log^2 m))$,*

$$
|\frac{1}{m}\langle Back_l(x_i) \cdot D_l, Back_{l'}(x_j) \cdot D'_{l'}\rangle| \le \mathcal{O}(\frac{L^4 \log^4 m}{m^{1/4}}).
\tag{26}
$$

*For $l = l'$, there is $F^l_{i,j}$ that, with probability at least $1 - exp(-\Omega(\log^2 m))$,*

$$
|\frac{1}{m}\langle Back_l(x_i) \cdot D_l, Back_l(x_j) \cdot D'_{l'}\rangle - F^l_{i,j}| \le \mathcal{O}(\frac{L^4 \log^4 m}{m^{1/4}}).
\tag{27}
$$

*where*

$$
\Sigma(x) = \frac{1}{2} + \frac{arcsin(x)}{\pi},
\tag{28}
$$

$$
F^l_{i,j} \succeq \frac{1}{K}\Sigma(\{\frac{1}{L^3}\langle X_{i,l}, X_{j,l}\rangle + K^{l-1}_{i,j}\}/Q_l).
\tag{29}
$$

*and $0 < K \le \mathcal{O}(1/L^4)$.*

**Remark 4.2** *We should note that this theorem is one of the key differences between this work and the methods in Allen-Zhu and Li (2019a). In fact, we must show that there is a constant $K > 0$ such that $\frac{1}{m}\langle Back_l(x_i), Back_l(x_j)\rangle - K$ is still positive definite. However, is $K$ large enough thus $1/K \geq poly(L)$ rather than $1/K \leq exp(-\Omega(L))$ ? This is not a trivial question. One can only get $K \geq \frac{1}{2^L}$ using naive estimation. In Allen-Zhu and Li (2019a), $||\boldsymbol{A}X_l|| \leq \epsilon_x$ is required to make sure $Back_l' = Back_l(x_i) - Back_l(x_j)$ samll. However after $k$ steps of training, we can show the approximation error is roughly $\mathcal{O}(||Back'||\cdot||\boldsymbol{W}^k - \boldsymbol{W}^0||)$ and $||\boldsymbol{W}^k - \boldsymbol{W}^0||_F \sim \sqrt{\widetilde{y}^T(\boldsymbol{H}^\infty)^{-1}\widetilde{y}} \sim \mathscr{C}(F^*)$. Thus the dependence of $\epsilon_x$ on $\mathscr{C}(F^*)$ is hard to be dealt with using this method. In this paper, we do not need the normalized condition. Our methods rely on a crucial observation that the function $\lim_{l\to\infty} h_l(x_i)^T h_l(x_j)/(||h_l(x_i)|| \cdot ||h_l(x_j||)$ will degenerate to a constant function.*

### 4.1.3 Sketch Proof of Theorem 4

In order to estimate the complexity, we use the results in the last subsection and Proposition 2.4,2.2 and 2.3.

Proposition 2.4 shows that, in order to estimate $\sqrt{\widetilde{y}^T(\boldsymbol{H}^\infty)^{-1}\widetilde{y}}$, we need to show

$$\boldsymbol{H}^\infty \succeq \xi_p \cdot (\boldsymbol{X}_l^T\boldsymbol{X}_l)^{\circ p} \tag{30}$$

with $\xi_p > 0$ for all $p \in \mathbb{N}, 1 \leq l \leq L$. Here $\boldsymbol{X}_l \in \mathbb{R}^{n\times d} = [X_{1,l}, X_{2,l}...X_{n,l}]$ and

$$[(\boldsymbol{X}_l^T\boldsymbol{X}_l)^{\circ p}]_{i,j} = \{X_{i,l}^TX_{j,l}\}^p. \tag{31}$$

We will show that, there is a matrix $H^\infty$. With probability at least $1 - \delta$, $H_{ij} = H_{ij}^\infty \pm \mathcal{O}(\frac{L^4 \log^4 m}{m^{1/4}})$ for all $i, j \in [n]$, and,

$$H_{i,j}^\infty \succeq \frac{1}{\mathcal{O}(L^4)} \cdot Q_l\Sigma(\{\frac{1}{L^3}\langle X_{i,l}, X_{j,l}\rangle + K_{i,j}^{l-1}\}/Q_l). \tag{32}$$

for all $l$.

Based on (32), we can show the following results:

For all $1 \leq l \leq L$ and all $k$

$$H_{i,j}^\infty \succeq \frac{1}{\mathcal{O}(L^4)}\Sigma(\{K_{i,j}^l + \frac{1}{L^3}X_{i,l}^TX_{j,l}\}/Q_l) \succeq \Omega(\frac{1}{L^7}) \cdot (\frac{1}{\mathcal{O}(L)})^k \cdot \frac{1}{k^2}(X_{i,l}^TX_{j,l})^k/(||X_{i,l}||\cdot X_{j,l}||)^k. \tag{33}$$

This deduces the complexity for the Additive Concept Class in section 2.1,

$$\sqrt{\widetilde{y}^T(\boldsymbol{H}^\infty)^{-1}\widetilde{y}} \leq \mathcal{O}(\mathscr{C}^*). \tag{34}$$

As for N-Variables Concept Class,

$$\begin{aligned}
H_{i,j}^\infty \succeq &\frac{1}{C_1^N L^4 \cdot L^{2N} \cdot C_{N,p} \cdot (p/N)^N} \\
&\cdot (X_{i,r_1}^TX_{j,r_1} + X_{i,r_2}^TX_{j,r_2}... + X_{i,r_N}^TX_{j,r_N})^p/(N \cdot \max_n(||X_{i,r_n}||) \cdot \max_n(||X_{j,r_n}||))^p
\end{aligned} \tag{35}$$

with some large constant $C_1 > 0$. Meanwhile, for any $l \leq L, a < l$, let $Z_{i,l,a} = [X_{i,l}, X_{i,l-1},...X_{i,l-a}]$. We have:

$$H_{i,j}^\infty \succeq \Omega(\frac{1}{L^7}) \cdot (\frac{1}{\mathcal{O}(L)})^k \cdot \frac{1}{k^2}(Z_{i,l,a}^TZ_{j,l,a})^k/(||Z_{i,l,a}|| \cdot Z_{j,l,a}|| \cdot 2^a)^k \tag{36}$$

Then from definition of complexity in section 2.2 and Proposition 2.4, we can prove

$$\sqrt{\widetilde{y}^T(\boldsymbol{H}^\infty)^{-1}\widetilde{y}} \leq \mathcal{O}(\mathscr{C}^*). \tag{37}$$

Therefore (20) follows.

# 5 Dissicusion

In this paper, we use a new method to avoid the normalized conditions. The main idea is to provide an esitmation for $\sqrt{\widetilde{y}^T(\boldsymbol{H}^\infty)^{-1}\widetilde{y}}$ in the RNN case directly. However, the value of $\sqrt{\widetilde{y}^T(\boldsymbol{H}^\infty)^{-1}\widetilde{y}}$ is only explicitly calculated for the two-layer case in Arora et al. (2019). In the RNN cases, the neural tangent kernel matrix involves the depth and the weight sharing in the network and difficult to deal with.

In Allen-Zhu and Li (2019a), their method is to reduce the RNN case to

$$f_L \approx \sum_l Back^{(0)} \cdot \mathbb{1}_{\langle W, h_{l-1}\rangle + AX_l \geq 0} W^* \cdot h_{l-1},$$

which is similar to a summation of $L$ two-layer networks. And this reduction requires the following operations in Allen-Zhu and Li (2019a):

1) Introduce new randomness to keep the independence of rows in the random initialization matrices W and A at different depths. Then estimate the perturbation.

2) Show the "off-target" Backward Correlation is zero.

3) Estimate the "on target" Backward Correlation by introducing a normalized input sequence $x^{(0)}$.

4) Explicitly construct the approximation.

These steps strongly rely on the normalized condition $||X_l|| \ll 1$ and this is apparently unrealistic. Instead, we calculate the kernel matrix and we introduce many new estimation to avoid this condition.

We should note that this expression

$$f_L \approx \sum_l Back^{(0)} \cdot \mathbb{1}_{\langle W, h_{l-1}\rangle + AX_l \geq 0} W^* \cdot h_{l-1}$$

is additive in itself. Thus the nonlinear interaction between different positions considered in this paper, especially N-variable target functions, **cannot be deduced** using the from this method. In the previous proof, Allen-Zhu and Li (2019a) is to use these steps to reduce the RNN function to a summation of two-layer networks and ignore the correlation between inputs from different locations and this heavily relies on the normalized condition. In our method, we need to consider the information in Back to show the non-linear correlation between the inputs at different positions and prove N-variable target functions are learnable, while Allen-Zhu and Li (2019a). requires the normalized condition to make sure $Back \approx Back^{(0)}$ to be roughly a constant. This is one of the most different parts between this work and Allen-Zhu and Li (2019a).

In our case, since we do no use the normalized condition, we must show the polynomial decay of the constant part in $Back$. As mentioned in Remark 4.2, in our case, it is generally non-trivial to show $\sqrt{\widetilde{y}^T(\boldsymbol{H}^\infty)^{-1}\widetilde{y}} \leq O(\mathscr{C}^*)$ with $\mathscr{C}^*$ polynomial in $L$. Our methods rely on a detailed estimation on the degeneracy of long RNN based on Theorem 5.

# 6 Related Work

**Overparameterized neural network.** In Tian (2017) and Du et al. (2018), it is shown that, for a single-hidden-node ReLU network, under mild assumptions, the loss function is one point convex in a very large area. However, in Safran and Shamir (2018), the authors pointed out that such good properties are rare for networks with multi-hidden nodes, and indicated that an over-parameterization assumption is necessary. Similarly, Hardt et al. (2016) showed that over-parameterization can help in the training process of a linear dynamic system i.e., linear RNN. A different way to show over-parameterization is important as in Freeman and Bruna (2016), this work proved that in the two-layer case if the number of the hidden nodes is large enough, the sub-level sets of the loss will be nearly connected. Their method can also be applied to deep networks with a skip connection in Wang et al. (2020) to study the properties of loss surfaces.

Recent breakthroughs were made in understanding the neural tangent kernel(NTK) Jacot et al. (2018); Alemohammad et al. (2021) of the neural network near the area of the random initialization. In Li and Liang (2018), Du et al. (2019), Allen-Zhu et al. (2019b) and Allen-Zhu et al. (2019c), it is shown

that deep networks with a large hidden size can attain zero training error, under some assumptions of input non-degeneracy. This explains the empirical results Zhang et al. (2017) that DNN can fit training data with even random labels.

There are also some provable convergence results with over-parameterization going beyond NTK. The loss surface of the two-layer over-parameterized network with quadratic activation function was studied in Du and Lee (2018) and Mahdi et al. (2018). They showed that all the bad local minima are eliminated by over-parameterization. For ReLU activation function, in Allen-Zhu and Li (2019b), it is shown that there exits some functions can not be learned by any kernel functions but learnable with less error by a network with a skip connection. Li et al. (2020) provided a convergence result for learning a specific two-layer neural network which can not be learned by any kernel method, including Neural Tangent Kernel.

### Generalization Ability of Deep Learning

Classical VC theory cannot explain the generalization ability of deep learning because the VC-dimension of neural networks is at least linear in the number of parameters Bartlett et al. (2019). Recently, Allen-Zhu et al. (2019a) showed that overparameterized neural networks can learn some notable concept classes of target functions with rich types. Moreover, their work goes beyond the NTK linearization and provides new results on the non-convex interactions of the three-layer network. Meanwhile, Arora et al. (2019) provided a fine-grained analysis on the generalization error and showed the connections to the matrix of the neural tangent kernel. The results were generalized to the multi-layer case in Cao and Gu (2019). Similar results were also studied in Ji and Telgarsky (2020) and Chen et al. (2020b).

Ref. Allen-Zhu et al. (2019a) also considered the generalization error bounds beyond the first-order NTK. It has been shown in Allen-Zhu et al. (2019a) that a three-layer ReLU network can provable learn some notable composite functions and dropout can help to reduce the Rademacher Complexity of the network thus reduce the generalization error bounds. The proof is based on the second-order NTK expansion and saddle points escaping arguments. Higher-order NTK are also studied in Bai and Lee (2020) with provable generalization error bounds. Moreover, it is shown in Chen et al. (2020a) that comparing with the general NTK, deep networks with neural representation can achieve improved sample complexities, while for the first-order NTK, depth may not provide benefits for the learning ability Bietti and Bach (2021).

## 7 Conclusion and Future Work

In this paper, we studied the problem of what type of function can be learned by RNN. In this work, we showed that RNNs can provably learn the two types of functions, the additive concept class and the N-variables concept class in *almost-polynomial in input length many iterations and samples* starting from random initialization. For the additive concept class, we proved the result without the normalized condition and showed the almost-polynomial complexity in input length $L$. For the N-variable concept class, we showed that RNN with ReLU activation function can provably learn functions like $\psi(\langle \beta, [X_{l_1}, ..., X_{l_N}] \rangle)$. The complexity of learning such functions grows exponentially with either $N$ or $l_0 = \max(l_1, ...l_N) - \min(l_1, ...l_N)$, but when one of them is small, the complexity is almost-polynomial in the input length $L$.

One of the limitations is that this work relies on the NTK linearization of RNN. One probably direction is to consider the non-convex interactions in RNN and learn more complex functions using the method in Allen-Zhu et al. (2019a). Meanwhile, this work studied RNN with ReLU activation function. This did not consider the "gate" structure in RNN. We believe that a study on GRU, LSTM, and MGU may lead to learning more complex functions with long-term memory.

## Acknowledgement

We would like to thank Professor Wenyu Zhang for his valuable discussion, and Shuai Wang for the great help in writing. We also thank the anonymous reviewers and area chair for their helpful comments. This research was funded by the Fundamental Research Funds for the Central Universities (Grant number 2020YJS012).

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
