# Supplementary Materials

## A  Flowchart of the Proofs

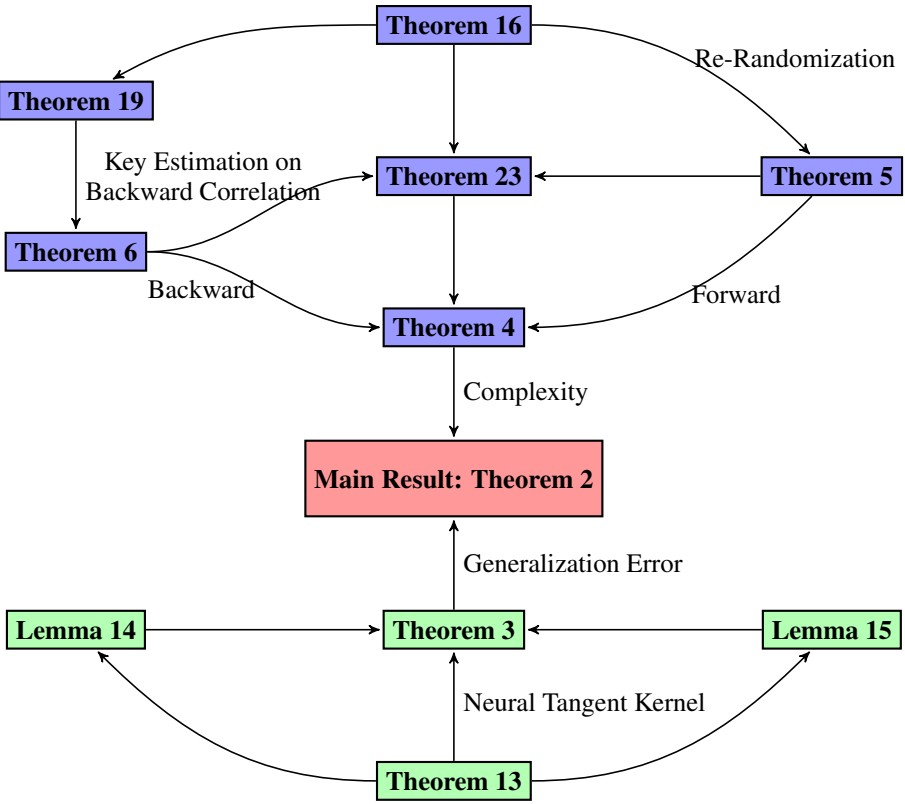

Figure 1: Flowchart of the Proof.

The Flowchart of the proof is shown in Figure 1. There are two parts. The first part is to prove Theorem 3. This is easy by using techniques in Allen-Zhu et al. (2019c) and Cao and Gu (2019). The more important part is to prove Theorem 4. We study the forward and backward correlation in Theorem 5 and 6. In Theorem 19, we show the polynomial degeneration of backward correlation which is crucial to show the complexity is polynomial in $L$.

## B  Some Probability Theory Lemmas

**Definition 1** *A random variable $X$ is said to be sub-Gaussian with variance proxy $\sigma^2$ if $\mathbb{E}[X] = 0$ and for all $s \in \mathbb{R}$,*

$$\mathbb{E}[e^{sX}] \leq e^{\frac{\sigma^2 s^2}{\sigma^2}}. \tag{38}$$

*A random variable $X$ is said to be $\lambda$-sub-exponential if $\mathbb{E}[X] = 0$, and for all $s$ that $|s| \leq \frac{1}{\lambda}$,*

$$\mathbb{E}[e^{sX}] \leq e^{\frac{s^2\lambda^2}{2}} \tag{39}$$

For $\lambda$-sub-exponential random variable, we have the following standard concentration inequality from Chernoff bound estimation(c.f. Boucheron et al. (2013)):

**Theorem 7** *Let $X_1, X_2, ...X_m$ be i.i.d $\lambda$-sub-exponential random variable with $\lambda < \mathcal{O}(1)$. Let $0 < \epsilon \leq 1$. With probability at least $1 - exp[\Omega(m\epsilon^2)]$,*

$$|\frac{1}{m}\sum_{i=1}^{m} X_i| \leq \epsilon \tag{40}$$

Let $\phi$ be a function with either $|\phi(x)| \leq |Bx|$ or $|\phi(x)| \leq B$ for some $B > 0$. Assuming $w$ is a Gaussian random vector, we can show $\phi(w^T X_1)\phi(w^T X_2) - \mathbb{E}\phi(w^T X_1)\phi(w^T X_2)$ is $\lambda$-sub-exponential for some $\lambda$ by estimating the moments. For $\mathbb{E}\phi(w^T X_1)\phi(w^T X_2)$, there is an equation which is a direct corollary of Claim 4.3 in Ge et al. (2017):

**Theorem 8** *Consider $M \in \mathbb{R}^d$, all the entries of $M$ are i.i.d. generated from $N(0, \frac{1}{d})$, and $X_1, X_2 \in \mathbb{R}^d$ with $||X_1|| = ||X_2|| = 1$. Let $\mu_i(\phi)$ denote the $i-th$ Hermite coefficient of function $\phi$, i.e. $\mu_r(\phi) = \frac{1}{\sqrt{2\pi}}\int_0^\infty \phi(x)h_r(x)e^{-\frac{x^2}{2}}dx$, $h_r(x) = \frac{1}{\sqrt{r!}}(-1)^r e^{\frac{x^2}{2}}\frac{d^r}{dx^r}e^{-\frac{x^2}{2}}$.*

*We have*

$$\mathbb{E}_M \phi_1(M^T X_2)\phi_2(M^T X_1) = \sum_r \mu_r(\phi_1)\mu_r(\phi_2)(X_1^T X_2)^r. \tag{41}$$

$$\mathbb{E}_M \phi(M^T X_2)\phi(M^T X_1) = \sum_r \mu_r^2(\phi)(X_1^T X_2)^r. \tag{42}$$

Combine the above two theorems and set $\epsilon = \frac{\log m}{\sqrt{m}}$. We have:

**Theorem 9** *Let $\boldsymbol{W} \in \mathbb{R}^{m \times d}$. All the entries of $M$ are i.i.d. generated from $N(0, \frac{2}{m})$, and $X_1, X_2 \in \mathbb{R}^d$ with $||X_1|| = ||X_2|| = 1$. $\phi(x) = \max(0, x)$ denotes the ReLU activation function. $\mu_i(\phi)$ denotes the $i-th$ Hermite coefficient of function $\phi$. $W_i$ denotes the i-th row of $\boldsymbol{W}$. With probability at least $1 - exp(-\Omega(\log^2 m))$,*

$$\phi^T(\boldsymbol{W}X_1)\phi(\boldsymbol{W}X_2) = \sum_i \phi^T(W_i X_1)\phi(W_i X_2)$$

$$= \mathbb{E}_{w \sim N(0, I_d)}\phi(w^T X_1)\phi(w^T X_2) \pm \mathcal{O}(\frac{\log m}{\sqrt{m}}) \tag{43}$$

$$= \sum_r \mu_r(\phi)\mu_r(\phi)(X_1^T X_2)^r \pm \mathcal{O}(\frac{\log m}{\sqrt{m}}).$$

This theorem is a direct corollary of the concentration inequality for the sub-exponential random variable $\phi(w^T X_1)\phi(w^T X_2)$.

In the case of ReLU function and its derivative, we can obtain analytical expressions which have been proved in Daniely et al. (2016); Huang et al. (2020):

**Theorem 10** *Consider functions $\phi_1(x) = \sqrt{2}\max(0, x)$ and $\phi_2(x) = \sqrt{2}\mathbb{1}\{x > 0\}$. Let $X_1, X_2 \in \mathbb{R}^d$, $||X_1|| = ||X_2|| = 1$, $z = X_1^T X_2$.*

$$\mathbb{E}_{w \sim N(0, I_d)}\phi_1(w^T X_1)\phi_1(w^T X_2) = \frac{\sqrt{1-z^2} + (\pi - arccos(z))z}{\pi}, \tag{44}$$

*and*

$$\mathbb{E}_{w \sim N(0, I_d)}\phi_2(w^T X_1)\phi_2(w^T X_2) = \frac{\pi - arccos(z)}{\pi}. \tag{45}$$

For such functions $f(z) = \mathbb{E}_{w \sim N(0, I_d)}\phi(w^T X_1)\phi(w^T X_2)$, we can see $f(0) = \mu_0^2(\phi)$ and $f'(0) = \mu_1^2(\phi)$.

## C   Technical Lemmas for RNN

Consider equations

$$h_l(\boldsymbol{W}, x) = \phi(\boldsymbol{W}h_{l-1} + \boldsymbol{A}X_l),$$
$$f(\boldsymbol{W}), x) = \boldsymbol{B}^T h_L(x),$$
$$\nabla f(\boldsymbol{W}, x_i) = \sum_{l=1}^L \text{Back}_l^T D_l \cdot h_l^T(x_i), \tag{46}$$
$$\text{Back}_l(\boldsymbol{W}, x_i) = \boldsymbol{B}^T D_L \boldsymbol{W} \cdots D_{l+1}\boldsymbol{W}.$$

The properties of $\nabla f(\boldsymbol{W}, x_i)$ and $h_l$ have been already appeared in Allen-Zhu et al. (2019c). We list the results we used in this section.

Let $\boldsymbol{W}_0$ be the point of Randomly Initialization and $B(\boldsymbol{W}_0, \omega) = \{\boldsymbol{W} | \,\, ||\boldsymbol{W} - \boldsymbol{W}_0||_F \leq \omega\}$. We have:

**Lemma 11** *For fixed vector $x \in \mathbb{R}^d$, $y, z \in \mathbb{R}^m$. With probability at least $1 - exp(-\Omega(m/L^2))$*

$$||\mathbb{1}_{\boldsymbol{W}_0 y + \boldsymbol{A} x > 0} \cdot \boldsymbol{W}_0 z||_2 \leq ||z||_2 (1 + 1/100L). \quad (47)$$

*For fixed $x \in \mathbb{R}^d$ and all $y, z$:*

$$||\mathbb{1}_{\boldsymbol{W}_0 y + \boldsymbol{A} x > 0} \cdot \boldsymbol{W}_0 z||_2 \leq ||z||_2 (1 + 1/50L). \quad (48)$$

The first equation is from Claim B.13 in Allen-Zhu et al. (2019c). The second one can be easily deduced from a $\epsilon$-net argument.

**Lemma 12** *(Section B and Section C in Allen-Zhu et al. (2019c)) Let $\tau_0 \leq poly(n, L)$, $\omega \leq \tau_0 \cdot m^{-1/2}$, $m \geq poly(L, n, \delta^{-1})$. With probability at least $1 - \delta$, for all $i \in [n]$, all $l$, and $\boldsymbol{W} \in B(\boldsymbol{W}_0, \omega)$*

*(a)* $||h_l(\boldsymbol{W}, x_i)|| \leq \mathcal{O}(l)$,

*(b)* $||Back_l(\boldsymbol{W}, x_i) D_l(\boldsymbol{W}, x_i)||_2 \leq \mathcal{O}(L^7 \sqrt{m})$,

*(c)* $||\boldsymbol{W} D_{l_1 - 1} ... \boldsymbol{W}(D_{l+1})|| \leq \mathcal{O}(L^7)$,

*(d)* *For any vector $v$ with $||v||_0 \leq \mathcal{O}(L^{10/3} \tau_0^{2/3} m^{2/3})$, $||B^T (D_L^0) \boldsymbol{W}^0 ... \boldsymbol{W}^0 v|| \leq \sqrt{m} L^{5/3} \tau_0^{1/3} \log m \cdot m^{-1/6}$,*

*(e)* $||D_l'||_0 \leq \mathcal{O}(L^{10/3} \tau_0^{2/3} m^{2/3})$.

The (a) is from the proof of Lemma B.3 and Lemma C.2a in Allen-Zhu et al. (2019c), and the (b) is from Lemma C.9 and Lemma B.11 in Allen-Zhu et al. (2019c). (c) is from Lemma C.7 in Allen-Zhu et al. (2019c). From Corollary B.18, Lemma C.11 and Claim G.2 in Allen-Zhu et al. (2019c) we have (d) and (e).

In our case, $||\boldsymbol{A} X_l|| \leq \frac{1}{L^{3/2}}$, rather than $||\boldsymbol{A} X_l|| \leq \mathcal{O}(1)$. These bounds can be improved, but since we mainly care about the exponential dependence on $L$, we do not use it.

These equations deduce the following linearization theorem which is an analogue of Lemma 4.1 in Cao and Gu (2019):

**Theorem 13** *With probability at least $1 - \mathcal{O}(n) \cdot exp(-\Omega(\log m))$, for all $i \in [n]$ and $\boldsymbol{W}, \boldsymbol{W}' \in B(\boldsymbol{W}_0, \omega)$,*

$$|f(\boldsymbol{W}', x_i) - f(\boldsymbol{W}, x_i) - \langle \nabla f(\boldsymbol{W}, x_i), \boldsymbol{W}' - \boldsymbol{W} \rangle| \leq \mathcal{O}(\omega^{1/3} L^{10} \log m \sqrt{m}) ||\boldsymbol{W}' - \boldsymbol{W}||_2. \quad (49)$$

**Proof:** Let
$$h_L(x) = h_L(\boldsymbol{W}, x), \,\, h_L(\boldsymbol{W}', x) = h_L(x) + h_L'(x),$$
$$D_l = D_l(\boldsymbol{W}, x), \,\, D_l' = D_l(\boldsymbol{W}', x), \,\, D_l^0 = D_l(\boldsymbol{W}_0, x). \quad (50)$$

By Claim G.2 in Allen-Zhu et al. (2019c), there exits diagonal matrices $D_l''$, $\{D_l''\}_{ii} \neq 0$ if and only if $\{D_l'\}_{ii} \neq 0$, $||D_l''||_0 \leq ||D_l'||_0 \leq \mathcal{O}(L^{10/3} \tau_0^{2/3} m^{2/3})$, and

$$B^T (h_L(x) + h_L'(x)) - B^T h_L(x) = \sum_{l=1}^{L-1} B^T (D_L + D_L'') \boldsymbol{W}' ... (D_{l+1} + D_{l+1}'') \quad (51)$$
$$\cdot (\boldsymbol{W}' - \boldsymbol{W}) h_l(x).$$

Then,
$$f(\boldsymbol{W}', x_i) - f(\boldsymbol{W}, x_i) - \langle \nabla f(\boldsymbol{W}, x_i), \boldsymbol{W}' - \boldsymbol{W} \rangle$$
$$= \sum_{l=1}^{L-1} B^T (D_L + D_L'') \boldsymbol{W}' ... (D_{l+1} + D_{l+1}'') \cdot (\boldsymbol{W}' - \boldsymbol{W}) h_l(x) \quad (52)$$
$$- B^T D_L \boldsymbol{W} ... \boldsymbol{W} D_{l+1} \cdot (\boldsymbol{W}' - \boldsymbol{W}) h_l(x).$$

To prove the theorem, same as Lemma 5.7 in Allen-Zhu et al. (2019b), we have the following result: Let $D_l^{0/1}$ be diagonal matrix and $(D_l^{0/1})_{ii} = 0$ if $(D_l + D_l'' - D_l^0)_{ii} = 0$, $(D_l^{0/1})_{ii} = 1$ if $(D_l + D_l'' - D_l^0)_{ii} \neq 0$. With probability at least $1 - \delta$,

$$||B^T(D_L + D_L'')\boldsymbol{W}'...(D_{l+1} + D_{l+1}'') \cdot \boldsymbol{W}') - B^T D_L \boldsymbol{W}...\boldsymbol{W} D_{l+1} \cdot \boldsymbol{W}||$$

$$\leq \mathcal{O}\Big( \sum_{l_1=l+1}^{L} \underbrace{||B^T(D_L^0)\boldsymbol{W}^0...\boldsymbol{W}^0(D_{l_1}^{0/1})||}_{T_1} \cdot ||D_{l_1}''|| \cdot \underbrace{||(D_{l_1}^{0/1})\boldsymbol{W}'D_{l_1-1}'...\boldsymbol{W}'(D_{l+1}')||}_{T_2}\Big) \quad (53)$$

$$\overset{(a)}{\leq} \mathcal{O}(\sqrt{m}L^{5/3+8}\tau_0^{1/3}\log m \cdot m^{-1/6}) \leq \mathcal{O}(\sqrt{m}L^{10}\omega^{1/3}\log m).$$

In (a), $T_2 \leq O(L^7)$ is from (c) in Lemma 12. From (d) in Lemma 12 and $||D_l''||_0 \leq \mathcal{O}(L^{10/3}\tau_0^{2/3}m^{2/3})$, $T_1 \leq \sqrt{m}L^{5/3}\tau_0^{1/3}\log m \cdot m^{-1/6}$. ∎

**Remark C.1** *In this theorem,*

$$|f(\boldsymbol{W}', x_i) - f(\boldsymbol{W}, x_i) - \langle \nabla f(\boldsymbol{W}, x_i), \boldsymbol{W}' - \boldsymbol{W} \rangle| \leq \mathcal{O}(\omega^{1/3}L^{10}\log m\sqrt{m})||\boldsymbol{W}' - \boldsymbol{W}||_2. \quad (54)$$

*And in Cao and Gu (2019), there is a similar result that*

$$|f(\boldsymbol{W}', x_i) - f(\boldsymbol{W}, x_i) - \langle \nabla f(\boldsymbol{W}, x_i), \boldsymbol{W}' - \boldsymbol{W} \rangle| \leq \mathcal{O}(\omega^{1/3}L^2\sqrt{\log m}\sqrt{m})||\boldsymbol{W}' - \boldsymbol{W}||_2. \quad (55)$$

*The differences on $\log m$ are from that Lemma 4.4 in Allen-Zhu et al. (2019b) says if $||u||_0 \leq s$, $|B^T(D_L + D_L'')\boldsymbol{W}_L'...(D_{l+1} + D_{l+1}'') \cdot \boldsymbol{W}_{l+1}'u| \leq \mathcal{O}(\sqrt{s\log m})$ and Corollary B.18 in Allen-Zhu et al. (2019c) says $|B^T(D_L + D_L'')\boldsymbol{W}'...(D_{l+1} + D_{l+1}'') \cdot \boldsymbol{W}'u| \leq \mathcal{O}(\sqrt{s}\log m)$ for RNN case.*

# D  Generalization properties: Proof of Theorem 3

**Lemma 14** *Denote $L_i(\boldsymbol{W}) = \ell(y_i \cdot f(\boldsymbol{W}, x_i))$. Suppose there exits $\boldsymbol{W}^* \in B(\boldsymbol{W}_0, R/\sqrt{m})$ with $R \leq poly(n, L)$, $L_i(\boldsymbol{W}^*) \leq \frac{1+R^2}{n}$. For any $\delta$, there exists*

$$m^*(n, \delta, R, L) = poly(n, R, L, \delta^{-1}) \quad (56)$$

*such that if $m > m^*$, with probability at least $1 - \delta$, SGD with $\eta = 1/m$ for some small enough $\nu$ will output:*

$$\frac{1}{n}\sum_{i=1}^{n} L_D^{0-1}(\boldsymbol{W}^i) \leq \mathcal{O}(\frac{1}{n}) + \mathcal{O}(\frac{R^2}{n}) + \mathcal{O}(\frac{\log(1/\delta)}{n}). \quad (57)$$

**Proof of Lemma 14:**

Firstly, for all $i$, $\boldsymbol{W} \in B(\boldsymbol{W}_0, \omega), \omega \leq R/m^{1/2}$, from Lemma 12, $||\nabla f(\boldsymbol{W}, x_i)||_F \leq \mathcal{O}(L^8\sqrt{m})$.

$$||\boldsymbol{W}^{i+1} - \boldsymbol{W}^0||_F \leq \sum_{k=1}^{i} ||\boldsymbol{W}^{k+1} - \boldsymbol{W}^k||_F \leq \mathcal{O}(n\eta L^8\sqrt{m}) \leq \frac{L^8 R}{\sqrt{m}} \leq \mathcal{O}(\tau_0/m^{1/2}) \quad (58)$$

with $\tau_0 \leq poly(n, L)$. Thus we can use Theorem 13. We have,

$$L_i(\boldsymbol{W}^i) - L_i(\boldsymbol{W}^*) \leq \langle \nabla_{\boldsymbol{W}} L_i(\boldsymbol{W}^i), \boldsymbol{W}^i - \boldsymbol{W}^* \rangle$$
$$+ |\ell'(y_i f(\boldsymbol{W}, x_i)) \cdot y_i| \cdot \mathcal{O}(\omega^{1/3}L^{10}\log m\sqrt{m})||\boldsymbol{W}^i - \boldsymbol{W}^*||_2$$
$$= \frac{\langle \boldsymbol{W}^i - \boldsymbol{W}^{i+1}, \boldsymbol{W}^i - \boldsymbol{W}^* \rangle}{\eta} + \mathcal{O}(\omega^{1/3}L^{10}\log m\sqrt{m})||\boldsymbol{W}^i - \boldsymbol{W}^*||_2$$
$$(59)$$

Therefore,

$$\sum_{i=1}^{n} L_i(\boldsymbol{W}^i) \leq \sum_{i=1}^{n}\{L_i(\boldsymbol{W}^*) + \frac{R^2}{2\eta m} + \mathcal{O}(\omega^{1/3}L^{10}\log m\sqrt{m})\sum_{i=1}^{n}||\boldsymbol{W}^i - \boldsymbol{W}^*||_2\},$$

$$\leq \sum_{i=1}^{n}\{L_i(\boldsymbol{W}^*) + \frac{R^2}{2\eta m} + \mathcal{O}(L^{10}\log m \cdot n \cdot R^{4/3} \cdot m^{-1/6})\}, \quad (60)$$

$$\overset{(a)}{\leq} \sum_{i=1}^{n} L_i(\boldsymbol{W}^*) + R^2.$$

In (a), we use $m > m^* \sim poly(n, L)$.

Therefore,

$$\frac{1}{n}\sum_{i=1}^{n} L_i(\boldsymbol{W}^i) \leq \frac{1+R^2}{n} + \frac{R^2}{n}. \tag{61}$$

The cross-entropy function $\ell(x)$ satisfies that $L_i'(\boldsymbol{W}^i) \leq L_i(\boldsymbol{W}^i)$ and $L_i^{0-1}(\boldsymbol{W}^i) \leq L_i'(\boldsymbol{W}^i)$, where

$$L_i'(\boldsymbol{W}^i) = -\ell'(y_i f(\boldsymbol{W}, x_i)). \tag{62}$$

And $-\ell'(x)$ is bounded. Using the boundedness and a martingale Bernstein bound argument as Lemma 4.3 in Ji and Telgarsky (2020), we have

$$\frac{1}{n}\sum_{i=1}^{n} L_D^{0-1}(\boldsymbol{W}^i) \leq \mathcal{O}(\frac{1}{n}) + \mathcal{O}(\frac{R^2}{n}) + \mathcal{O}(\frac{\log(1/\delta)}{n}). \tag{63}$$

∎

**Remark D.1** *The result of generalization error $1/n$ is this better than that in Cao and Gu (2019) $1/\sqrt{n}$, which shows*

$$\frac{1}{n}\sum_{i=1}^{n} L_D^{0-1}(\boldsymbol{W}^i) \leq \frac{4}{n}\sum_{i=1}^{n} L_i(\boldsymbol{W}^*) + \mathcal{O}(\frac{R}{\sqrt{n}}) + \mathcal{O}(\sqrt{\frac{\log(1/\delta)}{n}}). \tag{64}$$

*This is because Lemma 4.3 in Ji and Telgarsky (2020) makes use of the boundedness of $L_i(\boldsymbol{W})$. Thus it is applicable in this theorem. There is also a similar argument in Lemma 5.6 of Chen et al. (2020b).*

**Lemma 15** *Under the condition of Theorem 3, with probability at least $1 - \delta$, there exits $\boldsymbol{W}^* \in B(\boldsymbol{W}_0, R/\sqrt{m})$, such that $L_i(\boldsymbol{W}^*) \leq \frac{1+R^2}{n}$, $R \leq \widetilde{\mathcal{O}}(L\sqrt{\widetilde{y}^T(\boldsymbol{H}^\infty)^{-1}\widetilde{y}})$.*

**Proof of Lemma 15:**
Let $\boldsymbol{\epsilon}$ be the matrix in (18),

$$\boldsymbol{G} = m^{-1/2} \cdot (vec[\nabla f(\boldsymbol{W}^0, x_1)], vec[\nabla f(\boldsymbol{W}^0, x_2)], ...vec[\nabla f(\boldsymbol{W}^0, x_n)]) \in \mathbb{R}^{m^2 \times n}.$$

$$\boldsymbol{G} + \boldsymbol{\epsilon} = \boldsymbol{P}\boldsymbol{\Lambda}\boldsymbol{Q}^T. \tag{65}$$

is the singular value decomposition. Note that $m^2 \gg n$. We can set $\boldsymbol{\epsilon}^T\boldsymbol{G} = 0$ without changing $\boldsymbol{\epsilon}^T\boldsymbol{\epsilon}$.

With probability at least $1 - \delta$, for all $i \in [n]$, $|f(\boldsymbol{W}^0, x_i)| \leq \mathcal{O}(L\log(n/\delta))$. We assume $w^* = \boldsymbol{P}\boldsymbol{\Lambda}^{-1}\boldsymbol{Q}^T(B \cdot \widetilde{y})$, with $0 < |f(\boldsymbol{W}^0, x_i)| + \log\{1/[exp(n^{-1}) - 1]\} + 0.01 < B \leq \mathcal{O}(L\log(n/\delta))$ for all $i \in [n]$, then $||w^*||_2^2 \leq B^2\widetilde{y}^T(\boldsymbol{H} + \boldsymbol{\epsilon}^T\boldsymbol{\epsilon})^{-1}\widetilde{y}$. and $\boldsymbol{G}^Tw^* = B \cdot \widetilde{y} - \boldsymbol{\epsilon}^Tw^*$. Meanwhile, reshape $w^*$ as $\boldsymbol{W}^* \in \mathbb{R}^{m \times m}$, then we have

$$\langle \nabla f(\boldsymbol{W}^0, x_i), \boldsymbol{W}^* - \boldsymbol{W}^0 \rangle = B \cdot \widetilde{y}_i \pm ||\boldsymbol{\epsilon}||_F \cdot \sqrt{\widetilde{y}^T(\boldsymbol{H} + \boldsymbol{\epsilon})^{-1}\widetilde{y}} = B \cdot \widetilde{y}_i \pm 0.01. \tag{66}$$

Therefore $\boldsymbol{W}^* \in B(\boldsymbol{W}_0, \widetilde{O}(L\mathscr{C}^*/\sqrt{m}))$, and

$$\begin{aligned}
\ell(y_i \cdot (f(\boldsymbol{W}^*, x_i))) &\leq \ell(y_i \cdot \{f(\boldsymbol{W}^0, x_i) + \langle \nabla f(\boldsymbol{W}^0, x_i), \boldsymbol{W}^* - \boldsymbol{W}^0 \rangle\}) \\
&\quad + |\ell'(y_i f(\boldsymbol{W}, x_i)) \cdot y_i| \cdot \mathcal{O}(L^{10}\log m \cdot n \cdot R^{4/3} \cdot m^{-1/6}) \\
&\leq \ell(y_i \cdot \{f(\boldsymbol{W}^0, x_i) + \langle \nabla f(\boldsymbol{W}^0, x_i), \boldsymbol{W}^* - \boldsymbol{W}^0 \rangle\}) \\
&\quad + R^2/n \\
&\leq \ell(\log(1/[exp(n^{-1}) - 1])) + R^2/n, \\
&\leq n^{-1} + R^2/n.
\end{aligned} \tag{67}$$

Thus $L_i(\boldsymbol{W}^*) \leq \frac{1+R^2}{n}$. ∎

Then Theorem 3 follows from Lemma 14 and 15.

# E Forward Correlation: Proof of Theorem 5

**Theorem 16** *Consider equation $h'_l(x_1) = \phi(W_l h'_{l-1}(x_l) + A_l X_l)$, where the entries of $W^l$ and $A^l$ are i.i.d. generated from $N(0, \frac{2}{m})$ and $N(0, \frac{2}{L^3 m})$. $W^l$, $A^l$ and $W^{l'}$, $A^{l'}$ are independent when $l \neq l'$. With probability at lesat $1 - L^2 exp(-\Omega(\log^2 m))$. For all $1 < l \leq L$, we have*

$$|h_l^T(x) h_l(x') - {h'_l}^T(x) h'_l(x')| \leq \mathcal{O}(L^2 \log^2 m / \sqrt{m}) \tag{68}$$

*for $x, x' = x_1, x_2$.*

In order to prove the theorem, firstly we claim that

**Lemma 17** *Let $h_l(x) = \phi(W h_{l-1}(x) + A X_l)$. $\widetilde{h}_l(x) = \phi(\widetilde{W} \widetilde{h}_{l-1}(x) + \widetilde{A} X_l)$ is defined by $\widetilde{W}, \widetilde{A}$. $\widetilde{W}, \widetilde{A}$ and $W, A$ are i.i.d. Then for any $0 < l, l' < L$, with probability at least $1 - L^2 exp(-\Omega(\log^2 m))$,*

$$|h_l^T(x) h_{l'}(x') - \overline{h}_l^T(x) \overline{h}_{l'}(x')| \leq \mathcal{O}(l^2 \log^2 m / \sqrt{m}) \tag{69}$$

*where*

$$\overline{h}_l(x) = \phi(W \widetilde{h}_{l-1}(x) + A X_l)$$
$$\overline{h}_{l'}(x') = \phi(W \widetilde{h}_{l'-1}(x') + A X'_l)$$

**Proof of Theorem 16:**
In the case $l = 1$, $h_1(x) = \phi(W h_0 + A X_1)$.

From Theorem 9 we have, with probability at least $1 - exp(-\Omega(\log^2 m))$

$$h_1^T(x) h_1(x') = \mathbb{E} h_1^T(x) h_1(x') \pm \mathcal{O}(\log^2 m / m) = {h'_1}^T(x) h'_1(x') \pm \mathcal{O}(\log^2 m / \sqrt{m}) \tag{70}$$

The theorem is true.

Supposing the theorem is true for $l$, for $l + 1$, using Lemma 17

$$h_{l+1}^T(x) h_{1+1}(x') = \mathbb{E} \phi(W \widetilde{h}_l(x') + A X'_{l+1}) \phi(W \widetilde{h}_l(x) + A X_{l+1}) \pm \mathcal{O}(l^2 \log^2 m / \sqrt{m}) \tag{71}$$

$$W \widetilde{h}_l(x) + A X_{l+1} = \begin{bmatrix} W & L^{3/2} A \end{bmatrix} \cdot \begin{bmatrix} \widetilde{h}_l(x) \\ \frac{1}{L^{3/2}} X_{l+1} \end{bmatrix} = M \cdot z \tag{72}$$

Thus

$$\mathbb{E} \phi(W \widetilde{h}_l(x') + A X'_{l+1}) \phi(W \widetilde{h}_l(x) + A X_{l+1})$$
$$= \mathbb{E}_{w \sim N(0, \sqrt{2} I_{m+d})} \phi(w^T z) \phi(w^T z') \tag{73}$$
$$= {h'_{l+1}}^T(x) h'_{l+1}(x') \pm \mathcal{O}(l^2 \log^2 m / \sqrt{m})$$
$$h_{l+1}^T(x) h_{1+1}(x') = {h'_{1+1}}^T(x) h'_{1+1}(x') \pm \mathcal{O}((l+1)^{16} \log^2 m / \sqrt{m}).$$

∎

**Proof of Lemma 17:**
In the case $l = 1$, this is true from Theorem 9.

For $l > 1$, let $U_l \in \mathbb{R}^{m \times 2l}$ denote column orthonormal matrix using Gram-Schmidt as

$$U_l = GS(h_1(x_1), h_1(x_2), ... h_l(x_1), h_l(x_2)). \tag{74}$$

We can write

$$W h_l = W U_{l-1} U_{l-1}^T h_{l-1} + W(I - U_{l-1} U_{l-1}^T) h_l, \tag{75}$$

and

$$h_{l+1}(x) = \phi(W U_{l-1} U_{l-1}^T h_l + W(I - U_{l-1} U_{l-1}^T) h_l + A X_{l+1}) \tag{76}$$

Consider

$$
\begin{aligned}
h_{l+1}(x_2)^T h_{l+1}(x_1) =&\phi(WU_{l-1}U_{l-1}^T h_l(x_2) + W(I - U_{l-1}U_{l-2}^T)h_l(x_2) + Ax_{2,l+1})^T \\
&\cdot \phi(WU_{l-1}U_{l-1}^T h_l(x_1) + W(I - U_{l-1}U_{l-1}^T)h_l(x_1) + Ax_{1,l+1})
\end{aligned}
\tag{77}
$$

We write $z_1 = (I - U_{l-1}U_{l-1}^T)h_l(x_1)$, $z_2 = (I - U_{l-1}U_{l-1}^T)h_l(x_2)$. $q_1 = U_{l-1}^T h_l(x_1), q_2 = U_{l-1}^T h_l(x_2)$.

$$
z_2 = \frac{\langle z_1, z_2 \rangle z_1}{\|z_1\|^2} + (I - z_1 z_1^T / \|z_1\|^2)z_2
\tag{78}
$$

Then

$$
\begin{aligned}
h_{l+1}(x_2)^T h_{l+1}(x_1) =&\phi(WU_{l-1}q_2 + W\frac{\langle z_1, z_2 \rangle z_1}{\|z_1\|^2} + W(I - z_1 z_1^T / \|z_1\|^2)z_2 + Ax_{2,l+1})^T \\
&\cdot \phi(WU_{l-1}q_1 + Wz_1 + Ax_{1,l+1})
\end{aligned}
\tag{79}
$$

Thus

$$
\begin{aligned}
h_{l+1}(x_2)^T h_{l+1}(x_1) =&\phi([M_1 \quad M_2 \quad M_3 \quad M_4] \cdot \begin{bmatrix} q_2 \\ \frac{\langle z_1, z_2 \rangle}{\|z_1\|} \\ \|(I - z_1 z_1^T / \|z_1\|^2)z_2\| \\ x_{2,l+1} \end{bmatrix})^T \\
&\cdot \phi([M_1 \quad M_2 \quad M_3 \quad M_4] \cdot \begin{bmatrix} q_1 \\ \|z_1\| \\ 0 \\ x_{1,l+1} \end{bmatrix})
\end{aligned}
\tag{80}
$$

where

$$
[M_1 \quad M_2 \quad M_3 \quad M_4] = \begin{bmatrix} WU_{l-1} & Wz_1/\|z_1\| & W(I - z_1 z_1^T / \|z_1\|^2)z_2/\|(I - z_1 z_1^T / \|z_1\|^2)z_2\| & A \end{bmatrix}
\tag{81}
$$

Let

$$
\begin{aligned}
E_2 &= \begin{bmatrix} q_2 \\ \frac{\langle z_1, z_2 \rangle}{\|z_1\|} \\ \|(I - z_1 z_1^T / \|z_1\|^2)z_2\| \\ x_{2,l+1} \end{bmatrix}) \\
E_1 &= \begin{bmatrix} q_1 \\ \|z_1\| \\ 0 \\ x_{1,l+1} \end{bmatrix})
\end{aligned}
\tag{82}
$$

We have $E_2^T E_1 = h_l(x_2)^T h_l(x_1) + x_{2,l+1}^T x_{1,l+1}$.

In order to study $h_{l+1}(x_2)^T h_{l+1}(x_1)$, we can follow the method in the proof of Claim B.4 and Claim B.4 in Allen-Zhu et al. (2019c). Firstly fix $E_1, E_2$, with probability at least $1 - exp(-\Omega(\log^2 m))$,

$$
\begin{aligned}
&\phi([M_1 \quad M_2 \quad M_3 \quad M_4] \cdot E_2)^T \cdot \phi([M_1 \quad M_2 \quad M_3 \quad M_4] \cdot E_1) \\
&= \mathbb{E}_{M \sim \mathcal{N}(0,I)} \phi(ME_2)\phi(ME_1) \pm \mathcal{O}(l+1)^2 \frac{\log^2 m}{\sqrt{m}})
\end{aligned}
\tag{83}
$$

Then use $\epsilon - net$. We also have for any $E_1, E_2$,

$$
\begin{aligned}
&\phi([M_1 \quad M_2 \quad M_3 \quad M_4] \cdot E_2)^T \cdot \phi([M_1 \quad M_2 \quad M_3 \quad M_4] \cdot E_1) \\
&= \mathbb{E}_{M \sim \mathcal{N}(0,I)} \phi(ME_2)\phi(ME_1) \pm \mathcal{O}(l+1)^2 \frac{\log^2 m}{\sqrt{m}})
\end{aligned}
\tag{84}
$$

Thus

$$
\begin{aligned}
&h_{l'+1}(x')^T h_{l+1}(x) \\
&= \overline{h}_{l'+1}(x')^T \overline{h}_{l+1}(x) \pm \mathcal{O}((l+1)^2 \log^2 m / \sqrt{m}).
\end{aligned}
\tag{85}
$$

The theorem follows. ∎

Combing above theorems, Theorem 8,9 and 10, we have

**Lemma 18** *Let*

$$Q_l = \sqrt{(1 + \frac{1}{L^3}\sum_{k=1}^{l}||X_{i,k}||^2) \cdot (1 + \frac{1}{L^3}\sum_{k=1}^{l}||X_{j,k}||^2)}$$

$$\Gamma(z) = \frac{\sqrt{1-z^2} + (\pi - arccos(z))z}{\pi}$$

*There exits $K_{i,j}^l$ such that with probability at least $1 - L^2 exp(-\Omega(\log^2 m))$,*

$$|h_l^T(x_i)h_l(x_j) - K_{i,j}^l| \leq \mathcal{O}(\frac{l^{16}\log^2 m}{\sqrt{m}}). \tag{86}$$

*And*

$$K_{i,j}^1 = Q_1 \cdot \Gamma([1 + \frac{1}{L^3}X_{i,1}^T X_{j,1}]/Q_1)$$
$$K_{i,j}^l = Q_l \cdot \Gamma(\{\frac{1}{L^3}X_{i,l}^T X_{j,l} + K_{i,j}^{l-1}\}/Q_l)^r \tag{87}$$

Thus Theorem 5 follows.

## F    Backward Correlation: Proof of Theorem 6

**Theorem 19** *For $l \neq l'$, with probability at least $1 - L^2 exp(-\Omega(\log^2 m))$,*

$$|\frac{1}{m}\langle Back_l(x_i) \cdot D_l, Back_{l'}(x_j) \cdot D'_{l'}\rangle| \leq \mathcal{O}(\frac{L^4\log^4 m}{m^{1/4}}). \tag{88}$$

*For $l = l'$, with probability at least $1 - L^2 exp(-\Omega(\log^2 m))$,*

$$\frac{1}{m}\langle Back_l(x_i) \cdot D_l, Back_l(x_j) \cdot D'_l\rangle \succeq \Omega(1/L^4) \cdot \Sigma(\{\frac{1}{L^3}\langle X_{i,l}, X_{j,l}\rangle + K_{i,j}^{l-1}\}/Q_l) \pm \mathcal{O}(\frac{L^4\log^4 m}{m^{1/4}}). \tag{89}$$

**Proof of (88):**

The proof of (88) is almost a line-by-line copy of the proof in section C of Allen-Zhu and Li (2019a), but there are some minor differences.

Let $\zeta_1, ..., \zeta_m$ be a random orthonormal basis of $\mathbb{R}^m$. Then divide all the $m$ coordinates into $\sqrt{m}$ chunks $N_1, N_2, ...N_{m^{1/2}}$ of the size $N = \sqrt{m}$.

Define

$$z_{1,0} = D_l\zeta_1, z'_{1,0} = D'_l\zeta_1, \ ... \ z_{N,0} = D_l\zeta_N, z'_{N,0} = D'_l\zeta_N \tag{90}$$

and

$$z_{i,a} = D_{l+a}W \cdots D_{l+1}WD_l z_{i,1}$$
$$z'_{i,a} = D'_{l'+a}W \cdots D'_{l+1}WD'_{l'}z'_{i,1} \tag{91}$$

$$Z_{p,a} = GS(h_1, ..., h_{\max(l,l')}, z_{1,1}, ..., z_{N,1}, z'_{1,1}, ..., z'_{N,1}, ..., z_{1,a}, ..., z_{p,a}, z'_{1,a}, ..., z'_{p,a}) \tag{92}$$

We claim that, with probability at least $1 - L^2 exp(-\Omega(\log^2 m))$, for all $a$,

$$||Z_{p,a}^T z_{p,a}|| \leq \mathcal{O}(\frac{L^3\sqrt{N}\log^3 m}{\sqrt{m}}). \tag{93}$$

When $a = 0$,

$$Z_{p,0}^T z_{p,0} = Z_{p,0}^T D_l\zeta_1 \tag{94}$$

With probability at least $1 - exp(-\Omega(\log^2 m))$,

$$||Z_{p,0}^T z_{p,0}|| \leq \mathcal{O}(l \log m / \sqrt{m}). \tag{95}$$

For $a > 1$,

$$Z_{p,a+1}^T z_{p,a+1} = Z_{p,a+1}^T D_{l+a+1}(W(I - Z_{p,a+1}Z_{p,a+1}^T)z_{p,a} + WZ_{p,a+1}Z_{p,a+1}^T z_{p,a}), \tag{96}$$

$$||Z_{p,a+1}^T D_{l+a+1} W Z_{p,a+1} Z_{p,a+1}^T z_{p,a})|| \leq ||D_{l+a+1} W Z_{p,a+1} Z_{p,a+1}^T z_{p,a})||$$
$$\leq ||Z_{p,a+1}^T z_{p,a}||(1 + \frac{1}{50L}), \tag{97}$$

The last step is from Lemma 11.

And

$$||Z_{p,a+1}^T D_{l+a+1} W(I - Z_{p,a+1} Z_{p,a+1}^T) z_{p,a}|| \leq \mathcal{O}(\frac{(l+a)^3 \sqrt{N} \log^2 m}{\sqrt{m}}). \tag{98}$$

is because $W(I - Z_{p,a+1}Z_{p,a+1}^T)z_{p,a} \sim N(0, (2\boldsymbol{I}/m) \cdot ||(I - Z_{p,a+1}Z_{p,a+1}^T)z_{p,a}||^2)$.

This claim follows that,

$$\sum_{p \in [N]} \Xi_p = \sum_p B^T(I - Z_{p,a}Z_{p,a}^T)z_{p,a} \cdot B^T(I - Z_{p,a'}Z_{p,a'}^T)z'_{p,a'} \pm \mathcal{O}(m^{1/4}L^3 \log^4 m) \tag{99}$$

In the case $a \neq a'$, $(I - Z_{p,a}Z_{p,a}^T)z_{p,a}$ and $(I - Z_{p,a'}Z_{p,a'}^T)z'_{p,a'}$ are mutually orthogonal. With probability at least $1 - exp(-\Omega(\log^2 m))$,

$$|\sum_p B^T(I - Z_{p,a}Z_{p,a}^T)z_{p,a} \cdot B^T(I - Z_{p,a'}Z_{p,a'}^T)z'_{p,a'}| \leq \mathcal{O}(\log^4 m) \tag{100}$$

Thus

$$|\frac{1}{m}\langle \text{Back}_l(x_i) \cdot D_l, \text{Back}_{l'}(x_j) \cdot D'_{l'} \rangle| \leq \mathcal{O}(\frac{L^3 \log^4 m}{m^{1/4}}). \tag{101}$$

There are $\sqrt{m}$ chunks, thus with probability at least $1 - \sqrt{m}L^2 exp(-\Omega(\log^2 m)) = 1 - L^2 exp(-\Omega(\log^2 m))$. (88) follows. ∎

**Proof of (89):**

For any $a$, we have,

$$z_{p,a+1} = D_{l+a+1}(W(I - Z_{p,a+1}Z_{p,a+1}^T)z_{p,a} + WZ_{p,a+1}Z_{p,a+1}^T z_{p,a}) \tag{102}$$

Thus,

$$||z_{p,a+1} - D_{l+a+1}W(I - Z_{p,a+1}Z_{p,a+1}^T)z_{p,a}|| \leq \mathcal{O}(\frac{L^3\sqrt{N}\log^3 m}{\sqrt{m}}) \tag{103}$$

We know that $\frac{1}{m}\langle \text{Back}_l(x_i) \cdot D_l, \text{Back}_l(x_j) \cdot D'_l \rangle = \sum_{i=1}^{m^{1/2}} \Theta_i$, where

$$\Theta_i = \sum_{p \in [N_i]} \Xi_p = \sum_p B^T(I - Z_{p,a}Z_{p,a}^T)D_{l+a}W(I - Z_{p,a}Z_{p,a}^T)...D_{l+1}W(I - Z_{p,1}Z_{p,1}^T)z_{p,0}$$
$$\cdot B^T(I - Z_{p,a}Z_{p,a}^T)D'_{l+a}W(I - Z_{p,a}Z_{p,a}^T)...D'_{l+1}W(I - Z_{p,1}Z_{p,1}^T)z'_{p,0}$$
$$\pm \mathcal{O}(m^{1/4}L^3 \log^4 m) \tag{104}$$

Combine the facts :

- With probability at least $1 - exp(-\Omega(\log^2 m))$,

$$\sum_p B^T z_{p,a} \cdot B^T z'_{p,a} = \langle z_{p,a}, z'_{p,a} \rangle \pm \mathcal{O}(\frac{\sqrt{N}L^2 \log^2 m}{\sqrt{m}}). \tag{105}$$

- Let

$$D_l = \phi(\boldsymbol{W} h_{l-1}(x_l) + \boldsymbol{A} X_l)$$
$$\widetilde{D_l} = \phi(\boldsymbol{W} \widetilde{h}_{l-1}(x_l) + \boldsymbol{A} X_l)$$

(106)

where $\widetilde{h}$ is $h_l$ define by re-randomization in Lemma 17. Then $|\langle D'_l, D_l \rangle - \langle \widetilde{D'_l}, \widetilde{D_l} \rangle| \leq \mathcal{O}(\frac{L^2 \log^2 m}{m})$

-

$$||Z_{p,a}^T z_{p,a}|| \leq \mathcal{O}(\frac{L^3 \sqrt{N} \log^3 m}{\sqrt{m}}).$$

(107)

and Claim F.1. Let $Q_l = \sqrt{(1 + \frac{1}{L^3} \sum_{k=1}^{l} ||X_k||^2) \cdot (1 + \frac{1}{L^3} \sum_{k=1}^{l} ||X'_k||^2)}$. With probability at least $1 - exp(-\Omega(\log^2 m))$, we have

$$\begin{aligned}
\langle z_{p,a}, z'_{p,a} \rangle =& \langle (I - Z_{p,a-1} Z_{p,a-1}^T) z_{p,a-1}, (I - Z_{p,a-1} Z_{p,a-1}^T) z'_{p,a-1} \rangle \\
& \cdot \Sigma(\{h_{l+a-1}^T h'_{l+a-1} + \frac{1}{L^3} X_{l+a}^T X_{l+a}\}/Q_{l+a}) \pm \mathcal{O}(\frac{L^3 \sqrt{N} \log^3 m}{\sqrt{m}}), \\
=& \langle (z_{p,a-1}, z'_{p,a-1}) \cdot \Sigma(\langle h_{l+a-1}, h'_{l+a-1} \rangle/Q_{l+a} \\
& + \frac{1}{L^3} \langle X_{l+a}, X'_{l+a} \rangle/Q_{l+a}) \pm \mathcal{O}(\frac{L^3 \sqrt{N} \log^3 m}{\sqrt{m}}),
\end{aligned}$$

(108)

where

$$\Sigma(x) = \frac{1}{2} + \frac{arcsin(x)}{\pi} = \frac{\pi - arccos(x)}{\pi}.$$

(109)

In order to study the constant term in

$$\Sigma(\langle h_{l+a-1}, h'_{l+a-1} \rangle/Q_{l+a} + \frac{1}{L^3} \langle X_{l+a}, X'_{l+a} \rangle/Q_{l+a}),$$

we need to study $\langle h_{l+a-1}, h'_{l+a-1} \rangle$.

The constant term in $\langle h_{l+a-1}, h'_{l+a-1} \rangle/Q_{l+a}$ is the sequence (Lemma 18):

$$\begin{aligned}
K_l &= \Gamma(K_{l-1} \cdot Q_{l-1}/Q_l), \\
\Gamma(x) &= x + \frac{\sqrt{1-x^2} - arccos(x)x}{\pi}.
\end{aligned}$$

(110)

Note that $K_l > 0$ is convergent. Meanwhile, the sequence $K'_l$,

$$\begin{aligned}
0 &< K'_1 < 1, \\
K'_l &= \Gamma(K'_{l-1}),
\end{aligned}$$

(111)

is also convergent Huang et al. (2020). We have $\lim_{l \to \infty} K'_l = \lim_{l \to \infty} K_l = 1$. The aim of us is to show $\sum_{l=1}^{L} \sqrt{(1 - K_l)} \leq \mathcal{O}(\log L)$.

Let $e_l = 1 - K_l$. Claim F.3 and F.2 below show that $e_l \sim \frac{1}{l^2}$ and

$$\text{The constant term in } \{\prod_{l=1}^{L} \Sigma(\langle h_{l-1}, h'_{l-1} \rangle/Q_l + \frac{1}{L^3} \langle X_l, X'_l \rangle/Q_l)\} \geq \Omega(1/L^b). \quad (112)$$

and in this case, $b = 3 + \frac{\log^2 L}{L} \leq 4$. Then (89) follows. ∎

**Claim F.1** *Let $D$ and $D'$ be diagonal matrix satisfying*

$$\begin{aligned}
(D)_{k,k} &= \mathbb{1}\{\boldsymbol{W} Y + \boldsymbol{A} X > 0\}, \\
(D')_{k,k} &= \mathbb{1}\{\boldsymbol{W} Y' + \boldsymbol{A} X' > 0\}.
\end{aligned}$$

(113)

*If $\langle Y, Z \rangle, \langle Y, Z' \rangle = 0$,*

$$\begin{aligned}
\mathbb{E}_{\boldsymbol{W}, \boldsymbol{A}} \langle D \boldsymbol{W} Z, D' \boldsymbol{W} Z' \rangle =& Z^T Z' \cdot \mathbb{E}_{w \sim N(0, I_m), a \sim N(0, \frac{1}{L^3} I_d)} \langle \phi'([w, a]^T [Y, X]), \phi'([w, a]^T [Y', X']) \rangle \\
=& Z^T Z' \cdot \Sigma(\{Y^T Y' + X^T X'\}/(||Y|| \cdot ||Y'|| + ||X|| \cdot ||X'||))
\end{aligned}$$

(114)

*with $\phi'(x) = \sqrt{2} \mathbb{1}\{x > 0\}$.*

**Proof of Claim F.1:**
In fact,

$$\mathbb{E}_{\boldsymbol{W},\boldsymbol{A}}\langle D\boldsymbol{W}Z, D'\boldsymbol{W}Z'\rangle = \langle Z', \nabla_{Y'}\langle Z, \nabla_Y \mathbb{E}_{w\sim N(0,I_m), a\sim N(0,\frac{1}{L^3}I_d)}$$
$$\langle \phi([w,a]^T[Y,X]), \phi([w,a]^T[Y',X'])\rangle\rangle. \tag{115}$$

with $\phi(x) = \sqrt{2}\max(0,x)$. Then (114) is clearly a corollary of (115) and $\langle Y, Z\rangle, \langle Y, Z'\rangle = 0$.

**Claim F.2** *Supposing $K_l \sim cos[\pi(1-(\frac{l}{l+1})^b)]+\xi_l$, $\sum_{l=l_1}^{L}\sqrt{\xi_l}\leq \mathcal{O}(1)$, $b>0$,*

$$\prod_{l=1}^{L}\frac{\pi - arccos(K_l)}{\pi} \geq \Omega(exp(-b\log L)) \geq \Omega(L^{-b}) \tag{116}$$

**Proof:** We use the inequality,

$$\prod_{l=1}^{L}(1-\frac{b}{l}-\Omega(\sqrt{\xi_l})) \geq \Omega(exp(-\sum_{l=1}^{L}\frac{b}{l}-\Omega(\sqrt{\xi_l}))). \tag{117}$$

Meanwhile, for harmonic series,

$$\sum_{l=1}^{L}\frac{b}{l} = b\log L + b\gamma + O(1/L^2) \tag{118}$$

where $\gamma \approx 0.57721$ is the Euler- Mascheroni constant. Thus the claim follows. ∎

**Claim F.3** *Let $e_l$ satisfy*

$$e_l = \frac{Q_{l-1}}{Q_l}e_{l-1} + \frac{Q_l - Q_{l-1}}{Q_l}$$
$$-\frac{\sqrt{1-(1-\frac{Q_{l-1}}{Q_l}e_{l-1}-\frac{Q_l-Q_{l-1}}{Q_l})^2} - arccos(\frac{Q_{l-1}}{Q_l}e_{l-1}+\frac{Q_l-Q_{l-1}}{Q_l})(1-\frac{Q_{l-1}}{Q_l}e_{l-1}-\frac{Q_l-Q_{l-1}}{Q_l})}{\pi}. \tag{119}$$

*For $l, L$ large enough , we have $e_l \leq 1 - cos[\pi(1-(\frac{l}{l+1})^{3+\frac{\log^2 L}{L}})]+\xi_l$ and $\sum_{l=l_1}^{L}\sqrt{\xi_l}\leq \mathcal{O}(1)$.*

Before proving this claim, we cite the following lemma in the proof of Lemma 15 in Huang et al. (2020):

**Lemma 20** *Let*

$$z_l = 1 - cos[\pi(1-(\frac{l}{l+1})^{3+\frac{\log^2 L}{L}})]. \tag{120}$$

$$z_l \geq z_{l-1} - \frac{\sqrt{1-(1-z_{l-1})^2} - arccos(z_{l-1})(1-z_{l-1})}{\pi} + \frac{3\pi^2\log^2 L}{l^3 L} + \frac{20\pi^2}{2l^4} \tag{121}$$

**Proof of Claim F.3:** Firstly, note that from the assumption of $||X_l||$, we have

$$\frac{Q_l - Q_{l-1}}{Q_l} \leq \mathcal{O}(\frac{1}{L^3}).$$

We will show there exits $q_l = z_l + \xi_l$ such that

$$q_l \geq \frac{Q_{l-1}}{Q_l}q_{l-1} + \frac{Q_l - Q_{l-1}}{Q_l}$$
$$-\frac{\sqrt{1-(1-\frac{Q_{l-1}}{Q_l}q_{l-1}-\frac{Q_l-Q_{l-1}}{Q_l})^2} - arccos(\frac{Q_{l-1}}{Q_l}q_{l-1}+\frac{Q_l-Q_{l-1}}{Q_l})(1-\frac{Q_{l-1}}{Q_l}q_{l-1}-\frac{Q_l-Q_{l-1}}{Q_l})}{\pi}. \tag{122}$$

Then $e_l \leq q_l$. The theorem follows.

Let

$$\frac{Q_l - Q_{l-1}}{Q_l} = \epsilon_l z_{l-1},$$

$$q_{l-1} = (1 + \theta_l) z_{l-1},$$

$$(1 + \theta_{l+1}) = \frac{Q_l - Q_{l-1}}{Q_l}(1 + \theta_l) + \epsilon_l, \tag{123}$$

$$\theta_{l_0+1} = 0.$$

Since $z_l < 1$, $\theta_l > 0$. And

$$\frac{Q_{l-1}}{Q_l}q_{l-1} + \frac{Q_l - Q_{l-1}}{Q_l} = (\frac{Q_l - Q_{l-1}}{Q_l}(1 + \theta_l) + \epsilon_l)z_{l-1} = (1 + \theta_{l+1})z_{l-1}.$$

Using Lemma 20, since $\frac{\sqrt{1-z^2} - arccos(z)(1-z)}{\pi} \sim O(z^{3/2})$, we claim that

$$(1 + \theta_{l+1})z_l \geq (1 + \theta_{l+1})z_{l-1}$$
$$- \frac{\sqrt{1 - (1 - (1 + \theta_l)z_{l-1})^2} - arccos((1 + \theta_l)z_{l-1})(1 - (1 + \theta_l)z_{l-1})}{\pi} \tag{124}$$

This is because $\theta_l > 0$, $(1 + \theta_l)^{3/2} \geq (1 + \theta_{l+1})$. Then we have

$$-\frac{\sqrt{1 - (1 - \frac{Q_{l-1}}{Q_l}q_{l-1} - \frac{Q_l - Q_{l-1}}{Q_l})^2} - arccos(\frac{Q_{l-1}}{Q_l}q_{l-1} + \frac{Q_l - Q_{l-1}}{Q_l})(1 - \frac{Q_{l-1}}{Q_l}q_{l-1} - \frac{Q_l - Q_{l-1}}{Q_l})}{\pi}$$

$$\leq (1 + \theta_{l+1})(z_l - z_{l-1}). \tag{125}$$

Therefore,

$$\frac{Q_{l-1}}{Q_l}q_{l-1} + \frac{Q_l - Q_{l-1}}{Q_l} -$$

$$\frac{\sqrt{1 - (1 - \frac{Q_{l-1}}{Q_l}q_{l-1} - \frac{Q_l - Q_{l-1}}{Q_l})^2} - arccos(\frac{Q_{l-1}}{Q_l}q_{l-1} + \frac{Q_l - Q_{l-1}}{Q_l})(1 - \frac{Q_{l-1}}{Q_l}q_{l-1} - \frac{Q_l - Q_{l-1}}{Q_l})}{\pi}$$

$$\leq [\frac{Q_l - Q_{l-1}}{Q_l}](1 + \theta_l)z_{l-1} + \epsilon_l z_{l-1} + (1 + \theta_{l+1})z_l - (1 + \theta_{l+1})z_{l-1}$$

$$= [\frac{Q_l - Q_{l-1}}{Q_l}(1 + \theta_l) + \epsilon_l - (1 + \theta_{l+1})]z_{l-1} + [1 + \theta_{l+1}]z_l$$

$$= [1 + \theta_{l+1}]z_l$$

$$= q_l. \tag{126}$$

Since

$$(1 + \theta_{l+1}) = \frac{Q_l - Q_{l-1}}{Q_l}(1 + \theta_l) + \epsilon_l, \tag{127}$$

we can write

$$(1 + \theta_{l+1}) = 1 + \sum_{l'=l_0}^{l}\prod_{j=l'}^{l}\frac{Q_j - Q_{j-1}}{Q_j}\epsilon_{l'}. \tag{128}$$

Then

$$(1 + \theta_{l+1}) \leq 1 + \mathcal{O}(\sum_{l'=l_0}^{l}\epsilon_{l'}) \tag{129}$$

$$q_l = (1 + \theta_{l+1})z_l \leq z_l + \mathcal{O}(\sum_{l'=l_0}^{l}\epsilon_{l'}z_l)$$

$$\leq z_l + \mathcal{O}(\sum_{l'=l_0}^{l}\frac{Q_{l'} - Q_{l'-1}}{Q_{l'}}\frac{(l')^2}{l^2}) \tag{130}$$

$$\leq z_l + \mathcal{O}(\frac{l}{L^3})$$

Since

$$\sum_{l=1}^{L} \sqrt{\frac{l}{L^3}} \le \mathcal{O}(1), \tag{131}$$

the theorem follows. ∎

# G   Complexity of Functions: Proof of Theorem 4.

In this section, we give the detailed proof of Theorem 4.

**Lemma 21** *Let*

$$\Sigma(x) = \frac{1}{2} + \frac{arcsin(x)}{\pi}. \tag{132}$$

*If $||Z_i||, ||Z_j|| \le \mathcal{O}(1)$, $\mu > 1$,*

$$\Sigma(\{\mu + \frac{1}{L^3} Z_i^T Z_j\}/Q_l) \succeq \Omega(\frac{1}{L^3}) \cdot (\frac{1}{\mathcal{O}(L)})^k \cdot \frac{1}{k^2} (Z_i^T Z_j)^k / (||Z_i|| \cdot Z_j||)^k \tag{133}$$

**Proof:** From the Taylor formula, for all $p \in \mathbb{N}$,

$$\Sigma(Z_i^T Z_j) \succeq \sum_{p=1}^{\infty} \frac{(Z_i^T Z_j)^{2p-1}}{2\pi(2p-1)^2}. \tag{134}$$

And

$$\Sigma([\mu + \frac{1}{L^3} Z_i^T Z_j]/Q_l) \succeq \sum_{p=1}^{\infty} \frac{(\mu + \frac{1}{L^3} Z_i^T Z_j)^{2p-1}}{2\pi(2p-1)^2 \cdot Q_l^{2p-1}}. \tag{135}$$

For any $k \in \mathbb{N}$, the coefficient of $[Z_i^T Z_j/L^3]^k/$ in $\Sigma([\mu_0 + \frac{1}{L^3} Z_i^T Z_j]/Q_l)$ will be larger than $\frac{a_k}{Q_l^k}$ with

$$a_k = \sum_{2p-1>k}^{\infty} \frac{1}{2\pi(2p-1)^2} \cdot (\frac{\mu}{Q_l})^{2p-1-k} \cdot \frac{2p-1 \cdot (2p-2) \cdot ... \cdot (2p-k)}{k!} \tag{136}$$

Consider

$$b_k = \sum_{2p-1>k}^{\infty} (\frac{\mu}{Q_l})^{(2p-1-k-2)} \cdot \frac{2p-1 \cdot (2p-2) \cdot ... \cdot (2p-k+2))}{k!} \tag{137}$$

$b_k = \Omega((\frac{\mu}{Q_l})^2) \cdot a_k$. Let

$$f(x) = \frac{1}{1-x^2}. \tag{138}$$

Then

$$b_k \ge \Omega(|f^{(k-2)}(\frac{\mu}{Q_l})| \cdot \frac{1}{k!}$$
$$= \frac{(k-2)!}{2 \cdot k!}[\frac{1}{(1-\frac{\mu}{Q_l})^{k-1}} + \frac{(-1)^{k-2}}{(1+\frac{\mu}{Q_l})^{k-1}} \tag{139}$$
$$\ge \Omega(\frac{1}{(k-2) \cdot (k-3)} \cdot \frac{Q_l^{k-1}}{(Q_l-\mu)^{k-1}})$$

Thus the coefficient of $(Z_i^T Z_j/L^3)^k$ in $\Sigma([\mu_0 + Z_i^T Z_j]/Q_l)$ will be larger than

$$\Omega(\frac{1}{(k-2) \cdot (k-3)} \cdot \frac{Q_l^{-1}}{(Q_l-\mu)^{k-1}} \frac{Q_l^2}{\mu^2}) \ge \Omega(Q_l - \mu) \cdot \frac{1}{(Q_l-\mu)^k \cdot k^2}.$$

Since

$$0 < C_1 \le ||X_{l,i}||^2, ||X_{l,j}||^2 \le C_2,$$

$$||Z_i||^2 / \sum_{l=1}^{L} ||X_{i,l}||^2 \sim \frac{1}{L},$$

$$||Z_j||^2 / \sum_{l=1}^{L} ||X_{j,l}||^2 \sim \frac{1}{L} \tag{140}$$

and

$$Q_l = \sqrt{(1 + \frac{1}{L^3}\sum_{k=1}^{l}||X_{i,k}||^2) \cdot (1 + \frac{1}{L^3}\sum_{k=1}^{l}||X_{j,k}||^2)}. \tag{141}$$

We have

$$\Sigma(\{\mu + \frac{1}{L^3}Z_i^T Z_j\}/Q_l) \succeq \Omega(\frac{1}{L^3}) \cdot (\frac{1}{\mathcal{O}(L)})^k \cdot \frac{1}{k^2}(Z_i^T Z_j)^k/(||Z_i|| \cdot ||Z_j||)^k. \tag{142}$$

The claim follows. ∎

Using this lemma, note that we can write $[K_{i,j}^{l-1} + \frac{1}{L^3}X_{i,l}^T X_{j,l}]/Q_l = \frac{\mu + \frac{1}{L^3}X_{i,l}^T X_{j,l} + T_{i,j}}{Q_l}$ with $T_{i,j} \succeq 0$ where $\mu = 1$ is the constant term in $K_{i,j}^{l-1}$ and

$$[K_{i,j}^{l-1} + \frac{1}{L^3}X_{i,l}^T X_{j,l} + T_{i,j}]/Q_l \succeq \frac{\mu + \frac{1}{L^3}X_{i,l}^T X_{j,l}}{Q_l}.$$

We have the following lemma:

**Lemma 22** *Under the condition of Lemma 18, for any $k \in \mathbb{N}$,*

$$\Sigma(\{K_{i,j}^l + \frac{1}{L^3}X_{i,l}^T X_{j,l}\}/Q_l) \succeq \Omega(\frac{1}{L^3}) \cdot (\frac{1}{\mathcal{O}(L)})^k \cdot \frac{1}{k^2}(X_{i,l}^T X_{j,l})^k/(||X_{i,l}|| \cdot X_{j,l}||)^k. \tag{143}$$

Now we can prove Theorem 4.

**Theorem 23** *Assume there is $\delta \in [0, e^{-1}]$. Let $n$ samples in $\mathcal{D}$ be $\{x_i, y_i\}_{i=1}^n$. $\widetilde{y} = [F^*(x_1), F^*(x_2), ...F^*(x_n)]^T$. $F^*$ is a function belonging to the concept class (9) or (11) such that $y_i \cdot F^*(x_i) \geq 1$ for all $i$. There exits matrix $\boldsymbol{H}^\infty$ satisfying:*

$$\boldsymbol{H} + \boldsymbol{\epsilon}^T\boldsymbol{\epsilon} \succeq \boldsymbol{H}^\infty \text{ with } ||\boldsymbol{\epsilon}||_F \leq 0.01/\mathcal{O}(\mathscr{C}^*) \tag{144}$$

*and*

$$\sqrt{\widetilde{y}^T(\boldsymbol{H}^\infty)^{-1}\widetilde{y}} \leq \mathcal{O}(\mathscr{C}^*). \tag{145}$$

**Proof:**

Firstly, using the forward and backward correlation Theorem 5 and Theorem 6,

$$\frac{1}{m}\langle \text{Back}_l(x_i) \cdot D_l, \text{Back}_l(x_j) \cdot D_l'\rangle \succeq \frac{1}{\mathcal{O}(L^4)}\Sigma(\{\frac{1}{L^3}\langle X_{i,l}, X_{j,l}\rangle + K_{i,j}^{l-1}\}/Q_l) \pm \mathcal{O}(\frac{L^4 \log^4 m}{m^{1/4}}) \tag{146}$$

and

$$|\frac{1}{m}\langle \text{Back}_l(x_i) \cdot D_l, \text{Back}_{l'}(x_j) \cdot D_{l'}'\rangle| \leq \mathcal{O}(\frac{L^4 \log^4 m}{m^{1/4}}). \tag{147}$$

for $l \neq l'$.

Thus

$$H_{i,j} = \frac{1}{m}\langle \nabla f(\boldsymbol{W}, x_i), \nabla f(\boldsymbol{W}, x_j)\rangle = \sum_{l=1}^{L}\frac{1}{m}\langle, \text{Back}_l^T D_l \cdot h_l^T(x_i), \text{Back}_l^T D_l \cdot h_l^T(x_j)\rangle$$
$$\pm \mathcal{O}(\frac{L^6 \log^4 m}{m^{1/4}}) \tag{148}$$

The closure of multiplication Proposition 2.2 for positive definite function concludes there exits semi-positive define matrix $\boldsymbol{M}$

$$H_{i,j} + M_{i,j} \succeq \frac{1}{\mathcal{O}(L^4)}\Sigma(\{\frac{1}{L^3}\langle X_{i,l}, X_{j,l}\rangle + K_{i,j}^{l-1}\}/Q_l). \tag{149}$$

with $M_{i,j} \leq \mathcal{O}(\frac{L^6 \log^2 m}{m^{1/4}})$. Then $||\boldsymbol{M}||_F \leq n^2 \frac{L^6 \log^2 m}{m^{1/4}}$. $\boldsymbol{M}$ is semi-positive define, therefore there exits $\boldsymbol{\epsilon}^T \boldsymbol{\epsilon} = \boldsymbol{M}$, $||\boldsymbol{\epsilon}||_F \leq 0.01/\mathscr{C}^*$ by SVD and reshaping since $m > poly(n, \mathscr{C}^*)$. Meanwhile let

$$\boldsymbol{G} = m^{-1/2} \cdot (vec[\nabla f(\boldsymbol{W}^0, x_1)], vec[\nabla f(\boldsymbol{W}^0, x_2)], ...vec[\nabla f(\boldsymbol{W}^0, x_n)]) \in \mathbb{R}^{m^2 \times n}.$$

Since $m^2 \gg n$, we can set $\boldsymbol{\epsilon}$ satisfying $\boldsymbol{\epsilon}^T \boldsymbol{G} = 0$ without changing $\boldsymbol{\epsilon}^T \boldsymbol{\epsilon}$.

For a function $\psi(\beta_{l,r}^T X_l/||X_l||) = \sum_{p=1}^{\infty} c_p(\beta_{l,r}^T X_l/||X_l||)^p$ with $||\beta_{l,r}|| \leq 1$, let

$$y_p = [c_p(\beta_{l,r}^T X_{1,l}/||X_{1,l}||)^p, ...c_p(\beta_{l,r}^T X_{n,l}/||X_{n,l}||)^p] \in \mathbb{R}^n. \tag{150}$$

Using Proposition 2.4, if $H_{i,j}^{\infty} \succeq \xi_p(X_{l,i}^T X_{l,j}/(||X_{l,i}|| \cdot ||X_{l,j}||))^p$,

$$y_p^T (\boldsymbol{H}^{\infty})^{-1} y_p \leq \frac{c_p^2 ||\beta_{l,r}^T||^{2p}}{\xi_p}.$$

In our case, from Lemma 22,

$$\xi_p = \Omega(\frac{1}{L^4}) \cdot \Omega(\frac{1}{L^3}) \cdot (\frac{1}{\mathcal{O}(L)})^p \cdot \frac{1}{p^2} \tag{151}$$

Note that

$$y \overset{def}{=} \sum_{p=1}^{\infty} y_p = [\psi(\beta_{l,r}^T X_{1,l}/||X_{1,l}||), ...\psi(\beta_{l,r}^T X_{n,l}/||X_{n,l}||)] \in \mathbb{R}^n.$$

We have

$$\sqrt{y^T (\boldsymbol{H}^{\infty})^{-1} y} \leq \sum_p \sqrt{y_p^T (\boldsymbol{H}^{\infty})^{-1} y_p} \leq \sum_{p=1}^{\infty} \frac{c_p ||\beta_{l,r}^T||^p}{\xi_p}. \tag{152}$$

In our case ,

$$\frac{1}{\sqrt{\xi_p}} \leq O(L^{3.5}) \cdot (\mathcal{O}(\sqrt{L}))^p \cdot p. \tag{153}$$

We have

$$\sqrt{\widetilde{y}^T (\boldsymbol{H}^{\infty})^{-1} \widetilde{y}} \leq \mathcal{O}(\mathscr{C}^*) \tag{154}$$

for Additive Concept Class (9).

For N-variables Concept Class (11)

$$F^*(x) = \sum_r \psi_r(\langle \beta_r, [X_{l_1}, ..., X_{l_N}] \rangle / \sqrt{N} \max ||X_{l_n}||).$$

We rewrite it as

$$F^*(x) = \sum_r \psi_r(\langle \beta_r, [X_{l_{max}}, ..., X_{l_{max}-N'}] \rangle / \sqrt{N} \max ||X_{l_n}||$$

$$l_{max} = \max(l_1, .., l_N), N' = \max(l_1, .., l_N) - \min(l_1, .., l_N).$$

Finally we prove that $\sqrt{\widetilde{y}^T (\boldsymbol{H}^{\infty})^{-1} \widetilde{y}} \leq \mathcal{O}(L^4 \sum_r \mathscr{C}_N(\psi_r, \mathcal{O}(\sqrt{L}))$.

Based on the structure of $H^{\infty}$, we have

$$H_{i,j}^{\infty} \succeq \frac{1}{\mathcal{O}(L^4)} \cdot \Sigma(\{K_{i,j}^{l_1} + \frac{1}{L^3} X_{i,l_1}^T X_{j,l_1}\}/Q_{l_1}) \cdot ... \cdot \Sigma(\{K_{i,j}^{l_N} + \frac{1}{L^3} X_{i,l_N}^T X_{j,l_N}\}/Q_{l_N}).$$

Then we have the follow claim

**Claim G.1** *For any N terms* $X_{i,r_1}^T X_{j,r_1}, X_{i,r_2}^T X_{j,r_2}..., X_{i,r_N}^T X_{j,r_N}$, $r_{max} = \max(r_1, ...r_N)$, *we have*

$$H_{i,j}^{\infty} \succeq \frac{1}{C_1^N L^4 \cdot L^{2N} \cdot C_{N,p} \cdot (p/N)^{2N}}$$

$$\cdot (X_{i,r_1}^T X_{j,r_1}/||X_{i,r_1}|| \cdot ||X_{j,r_1}|| + X_{i,r_2}^T X_{j,r_2}/||X_{i,r_2}|| \cdot ||X_{j,r_2}||...$$

$$+ X_{i,r_N}^T X_{j,r_N}/||X_{i,r_N}|| \cdot ||X_{j,r_N}||)^p$$

$$\succeq \frac{1}{C_1^N L^4 \cdot L^{2N} \cdot C_{N,p} \cdot (p/N)^{2N}}$$

$$\cdot (X_{i,r_1}^T X_{j,r_1} + X_{i,r_2}^T X_{j,r_2}... + X_{i,r_N}^T X_{j,r_N})^p/(N \cdot \max_n(||X_{i,r_n}||) \cdot \max_n(||X_{j,r_n}||))^p \tag{155}$$

*where $C_1$ is a large constant.*

which can be deduced from the following facts:

(a) For $k \in \mathbb{N}$, $\Sigma(\{K_{i,j}^l + \frac{1}{L^3}X_{i,l}^T X_{j,l}\}/Q_l) \succeq \Omega(\frac{1}{L^3}) \cdot (\frac{1}{\mathcal{O}(L)})^k \cdot \frac{1}{k^2}(X_{i,l}^T X_{j,l})^k/(||X_{i,l}|| \cdot X_{j,l}||)^k$.

(b) For any $n$ integers $n_1, n_2, ...n_N$, with $n_1 + n_2 + .. + n_N = p$, $C_{N,p} \geq \frac{p!}{n_1!n_2!...n_N!}$ and the largest coefficient of monomial in $(x_1 + x_2+, .. + x_N)^{2p-1}$ is less than $C_{N,p}$.

(c) For any $n$ integers $n_1, n_2, ...n_N$, with $n_1 + n_2 + .. + n_N = p$, $(p/N)^{2N} \geq n_1^2 \cdot ... \cdot n_N^2)$.

(b) and (c) are trivial. (a) is from Lemma 22.

Combing these results, polynomial theorem and using a similar argument as (154), we have

$$\sqrt{\widetilde{y}^T(\boldsymbol{H}^\infty)^{-1}\widetilde{y}} \leq L^2\mathcal{O}(1 + \sum_{p=1}^\infty L^{1.5N}C_1^N \cdot \sqrt{C_{N,p}} \cdot (p/N)^N(\mathcal{O}(\sqrt{L}))^p \cdot |c_p|.$$

Thus $\sqrt{\widetilde{y}^T(\boldsymbol{H}^\infty)^{-1}\widetilde{y}} \leq \mathcal{O}(L^2\sum_r \mathscr{C}_N(\psi_r, 1))$.

Finally we prove

$$\sqrt{\widetilde{y}^T(\boldsymbol{H}^\infty)^{-1}\widetilde{y}} \leq \mathcal{O}(L^3\sum_r \mathscr{C}(\psi_r, 2^{l_0}\mathcal{O}(\sqrt{L}))).$$

Consider

$$K_{i,j}^1 = Q_l \cdot \sum_{r=0}^\infty \mu_r^2(1 + \frac{1}{L^3}X_{i,1}^T X_{j,1}/Q_l)^r,$$

$$K_{i,j}^l = Q_l \sum_{r=0}^\infty \mu_r^2(\{\frac{1}{L^3}X_{i,l}^T X_{j,l} + K_{i,j}^{l-1}\}/Q_l)^r. \tag{156}$$

with $\mu_r = \frac{1}{\sqrt{2\pi}}\int_0^\infty \sqrt{2}xh_r(x)e^{-\frac{x^2}{2}}dx$, $h_r(x) = \frac{1}{\sqrt{r!}}(-1)^re^{\frac{x^2}{2}}\frac{d^r}{dx^r}e^{-\frac{x^2}{2}}$. We can rewrite this equation as:

$$\overline{K}_1 = \sum_{r=0}^\infty \mu_r^2(1 + \frac{1}{L^3}X_{i,1}^T X_{j,1}/Q_1)^r = \Gamma([1 + \frac{1}{L^3}X_{i,1}^T X_{j,1}]/Q_1),$$

$$\overline{K}_l = \Gamma(\overline{K}_{l-1} \cdot Q_{l-1}/Q_l), \tag{157}$$

$$\Gamma(x) = x + \frac{\sqrt{1-x^2} - arccos(x)x}{\pi}.$$

and

$$K_{i,j}^l = \overline{K}_l \cdot Q_l$$
$$= Q_l \cdot \Gamma\underbrace{\frac{Q_{l-1}}{Q_l} \circ ... \circ \Gamma\{\frac{1}{Q_1} \cdot (1 + \frac{1}{L^3}X_{i,1}^T X_{j,1})\}}_{l\ times}. \tag{158}$$

Using the fact

$$Q_l \cdot \prod_{k=k_0}^l \frac{Q_{l-1}}{Q_l} = Q_{k_0-1}, \tag{159}$$

and

$$\nabla_x f_l \circ f_{l-1}... \circ f_1(x) = f_l' \circ f_{l-1}'... \circ f_1'(x),$$

The linear part in $K_{i,j}^l$ is $\sum_{r=0}^{l-1}\mu_1^{2l-2r}\frac{1}{L^3}X_{i,r}^T X_{j,r}$. Thus

$$K_{i,j}^l + \frac{1}{L^3}X_{i,l}^T X_{j,l}$$

$$\succeq \frac{1}{L^3}X_{i,l}^T X_{j,l} + \sum_{r=0}^{l-1}\mu_1^{2l-2r}\frac{1}{L^3}X_{i,r}^T X_{j,r} \tag{160}$$

$$\succeq \mu_1^{2l}\sum_{r=1}^l \frac{1}{L^3}X_{i,r}^T X_{j,r}$$

with $\mu_1^2 = \frac{1}{2}$.

$||\mu_1^{2l} \sum_{r=1}^{l} X_{i,r}^T X_{j,r}|| \leq \mu_1^{2l} \cdot l \leq \mathcal{O}(1)$. Then from Lemma 21, we have

$$H_{i,j}^{\infty} \succeq \Omega(\frac{1}{L^7}) \cdot (\frac{1}{\mathcal{O}(L)})^k \cdot \frac{1}{k^2} (Z_i^T Z_j)^k / (||Z_i|| \cdot ||Z_j|| \cdot 2^l)^k \qquad (161)$$

with $Z_i^T Z_j = \sum_{r=1}^{l} \mu_1^{2r} X_{i,r}^T X_{j,r}$ and $||Z_i||^2 = \sum_r \mu_1^{2r} ||X_{i,r}||^2$

Therefore

$$\sqrt{\widetilde{y}^T (\boldsymbol{H}^{\infty})^{-1} \widetilde{y}} \leq \mathcal{O}(L^{3.5} \sum_r \mathscr{C}(\psi_r, 2^{l_0} \mathcal{O}(\sqrt{L})))$$

The theorem follows. $\qquad\qquad\qquad\qquad\qquad\qquad\qquad\qquad\qquad\qquad\qquad\qquad\blacksquare$

**Remark G.1** *Based on the previous results, we can generalize the results to the loss with the form:*

$$\frac{1}{n} \sum_{i=1}^{n} \sum_{l=1}^{L} \ell(y_i \cdot f_l(\boldsymbol{W}, x_i))$$

*with $f_l(\boldsymbol{W}, x) = \boldsymbol{B}^T h_l(x)$ to show for $\boldsymbol{H}_{i,j}^l = \langle \nabla f_l(\boldsymbol{W}, x_i), \nabla f_l(\boldsymbol{W}, x_j) \rangle$, there exits*

$$\boldsymbol{H}^l + \boldsymbol{\epsilon}^T \boldsymbol{\epsilon} \succeq (\boldsymbol{H}^l)^{\infty} \text{ with } ||\boldsymbol{\epsilon}||_F \leq 0.01/\sqrt{\widetilde{y}^T ((\boldsymbol{H}^l)^{\infty})^{-1} \widetilde{y}} \qquad (162)$$

*In fact we have following two generalization results of previous results which are in fact already contained in the proof.*

*Generalization of Lemma 17:*

*Let $g_l = \phi_1(\boldsymbol{W} g_{l-1})$, $h_l(x_1) = \phi_2(\boldsymbol{W} h_{l-1}(x_l) + \boldsymbol{A} X_l)$. $\widetilde{g}_l = \phi_1(\widetilde{\boldsymbol{W}} \widetilde{g}_{l-1})$ and $\widetilde{h}_l(x_1) = \phi_1(\widetilde{\boldsymbol{W}} \widetilde{h}_{l-1}(x_l) + \widetilde{\boldsymbol{A}} X_l)$ are defined by $\widetilde{\boldsymbol{W}}, \widetilde{\boldsymbol{A}}$. $\widetilde{\boldsymbol{W}}, \widetilde{\boldsymbol{A}}$ and $\boldsymbol{W}, \boldsymbol{A}$ are i.i.d. Then for any $0 < l, l' < L$, with probability at least $1 - L^2 exp(-\Omega(\log^2 m))$,*

$$|g_l^T h_{l'}(x') - \overline{g}_l^T \overline{h}_{l'}(x')| \leq \mathcal{O}(L^2 \log^2 m/m) \qquad (163)$$

*where*

$$\overline{g}_l = \phi_1(\boldsymbol{W} \widetilde{g}_{l-1})$$
$$\overline{h}_{l'}(x_1) = \phi_2(\boldsymbol{W} \widetilde{h}_{l'-1}(x_l) + \boldsymbol{A} X_l)$$

*Let $\phi_1(x) = x, \phi_2(x) = max(x, 0)$. One corollary of this result is that from (4.2) in Allen-Zhu et al. (2019c), there exits $g_l$, such that $\langle g_l, h_{l'} \rangle \geq 1/poly(L)$ when $l = l'$. Else $\langle g_l, h_{l'} \rangle = 0$.*

*Generalization of Theorem 19:*

*With probabiluty at least $1 - L^2 exp(-\Omega(\log^2 m))$,*

$$|\frac{1}{m} \langle B D_{l_1} W \cdots D_{l+1} W D_l, B D'_{l_2} W \cdots D'_{l+1} W D'_l \rangle| \leq \mathcal{O}(\frac{L^4 \log^4 m}{m^{1/4}}). \qquad (164)$$

*if $l_1 \neq l_2$.*

*Then we can show there exits $w_a^*$ with $||w_a^*|| \leq \mathscr{C}(F_a^*)$. for $a = 1, 2...L$ with*

$$\frac{1}{\sqrt{m}} \langle \nabla_{\widetilde{W}} f_a(\widetilde{\boldsymbol{W}}, x_i), w_a^* \rangle = \frac{1}{m} \sum_l \langle B D_a W \cdots D_{l+1} W D_l, w_{a,back}^* \rangle \cdot \langle h_l(x_i), g_a \rangle = F_a^*(x_i) + \epsilon$$
$$(165)$$

*and*

$$|\frac{1}{\sqrt{m}} \langle \nabla_{\widetilde{W}} f_{a'}(\widetilde{\boldsymbol{W}}, x_i), w_a^* \rangle = |\frac{1}{m} \sum_l \langle B D_{a'} W \cdots D_{l+1} W D_l, w_{a,back}^* \rangle \cdot \langle h_l(x_i), g_a \rangle| \leq \epsilon \quad (166)$$

*when $a \neq a'$.*

*Here $w_{a,back}^*$ is from the SVD of matrix*

$$\frac{1}{m} \langle B D_{l_1} W \cdots D_{l+1} W D_l, B D'_{l_2} W \cdots D'_{l+1} W D'_l \rangle.$$

*as (65) in the proof of Lemma 15.*