# OpenReview forum: "On the Provable Generalization of Recurrent Neural Networks"
_NeurIPS.cc/2021/Conference — NeurIPS 2021 Poster_

### Official Review · Reviewer_yR1a · 2021-07-14

**Rating:** 5
**Confidence:** 3

**Summary:**

This paper investigates the learnability of certain types of concept classes via recurrent neural networks. In particular, the authors consider two concept classes: 1) functions that are sums of power series of a linear function of each temporal input and 2) functions that are sums of power series of a linear transformation of the entire temporal sequence. Under suitable conditions, it is proved that these functions can be learned by RNN in polynomial time and sample complexity, and almost polynomial in input sequence length.

**Main Review:**

While the topic of learnability of function(al)s by RNN is a highly relevant topic, I am not sure if the results in this paper contributes to a substantial advancement in this understanding. The biggest issue I see is the lack of motivation of why one would want to consider concept classes introduced in Eq (4) and (5), and in particular why they are interesting generalizations of those considered in earlier works (e.g. [Allen-Zhu and Li, 2019a]). Moreover, the proof techniques appears to be adapted from earlier works. In this sense, I feel that the novelty of the results presented here is quite limited. Below, I list some additional questions/issues:

1. As far as I am aware, ReLU activation is rarely used in RNNs as a recurrent activation, since it only outputs non-negative values. Usually, activations such as $tanh$ is used. How crucially do the current results depend on this assumption?
2. What is the intuition of the complexity measures in (6) and (7), and what is the relationship between usual Hardy/Bergman spaces and their associated associated norms? It appears to me to be a measure of (generalized) regularity, since it concerns the decay rate of certain series coefficients. The paper would benefit from a discussion of the meaning of the complexity measure introduced.
3. Algorithm 1: As stated, the matrices $A$ and $B$ appears to be fixed during training. This is different from practical RNN training where these are also weights to be trained.

## Minor Issues

The paper would benefit from more careful proofreading, as there are many typos and grammatical errors. I list some I have identified below.

1. Page 5 Line 155: the parameters of $m^*$ does not agree with previous line.
2. Page 5 Line 155: “there exits”
3. Page 6 Line Remark 3.3: “do not dependent”
4. Page 6 Line 180: “The really matter thing” - revise sentence
5. Eq (4): and Eq (5): the RHS should depend on $l,r$ and $r$ respectively

---

After rebuttal: I thank the authors for addressing some of the questions. It appears that some of the limitations I pointed out (not training A,B, using ReLU activations) are also assumed in some prior works. While still unrealistic, it is acceptable for the purpose of simplifying analysis to obtain insights.
I think the paper's novelty can be much better emphasized by building some of the discussion in the authors' reply in the paper itself. In the present form I have reservations to recommend acceptance due to unclear new contributions.

**Time Spent Reviewing:**

4

---

> ### Author Response · Authors · 2021-08-06
> **Response to Reviewer yR1a**
>
>
> We thank the reviewer for the review and support. We are particularly grateful to the reviewer to  for catching the missing in expressing  the motivation and algorithm setup.
>
>
> **Responses to the motivation:**
>
> In this paper, our motivation  is to study what kind of functions are learnable for RNNs. We generalize the previous works  in the following two directions:
>
> Case (a)  Additive Concept Class in  (4) $\sum_l \psi_l (\beta_l^T X_l)$
>
> This is the same as the target functions in [Allen-Zhu and Li, 2019a], but we do not need normalized conditions in [Allen-Zhu and Li, 2019a].
>
> Case (b)  N-variables Concept Class in (5) $\sum_r \psi_r (\beta_r^T [X_{l_1}, X_{l_2},... X_{l_N}])$
>
> This is new.
>
> All the results in [Allen-Zhu and Li, 2019a] are to show the case (a) is learnable. However,  the results in  [Allen-Zhu and Li, 2019a] required  the normalized condition $||X_l||\leq \epsilon^3/\mathscr{C}^3$, to achieve the  $O(\epsilon)$ generalization error bound , where $\mathscr{C}$ is the complexity of the targe function.  The normalized condition is crucial in their proof but it is obviously unnatural and unrealistic since in practice, $\mathscr{C}$ can be very large but the norm of the input will not be that small. One of our main purposes is to study the learning ability of RNN without this normalized condition.
>
>
> In the case (b), we study the N-variables Concept Class. As an example,  we consider the  target functions with the form $\psi(\beta^T [X_1, X_L])$. We think this generalization is a naturally valuable because from Hahn-Banach theorem (or equivalent, the universal approximation theorem of neural networks with the activation function $\psi$), there exits  a suitable function $\psi$ such that $\psi(\beta^T_r [X_1, X_L])$ with different $r$ form a complete basis on the space  of all the continuous functions with input $X_1, X_L$. Thus $\sum_r c_r \psi(\beta^T_r [X_1, X_L])$ is a large enough kind of functions. The functions in the case (a) only consider the additive interaction while case (b) contains the non-linear functions of two variables $X_1$ and $X_L$ which can not be written as $f(X_1)+g(X_L)$.
>
> **Responses to the novelty:**
>
> We apologize that the original representation was not clear enough to explain the novelty.
>
> As shown in Figure 1, there are two parts in the proof. The proof techniques to prove Theorem 3 are indeed adapted from earlier works, especially [Cao and Gu, 2019]. However, the main step in this paper is to prove Theorem 4, i.e. Eq (20) $\sqrt{\widetilde{y}^T (H^\infty)^{-1}\widetilde{y}}\leq O(\mathscr{C}^*)$.  In the supplementary material, pages 18 to 30 devote to prove Theorem 4 and only the proof of (84) is adapted from the earlier work [Allen-Zhu and Li, 2019a].
>
> Although the problem studied in this paper is the same as that in [Allen-Zhu and Li, 2019a],  the framework in [Allen-Zhu and Li, 2019a] is from the viewpoint of constructing an approximation. Yet we use a totally different but more powerful method in this paper. The generalizations considered in this paper, especially N-variable target functions, are very hard to be proved using the previous approximation-based methods. This is because the main idea in the previous proof in [Allen-Zhu and Li, 2019a] is to reduce the RNN function to a summation of two-layer networks as $f_L\approx\sum_lBack^{(0)} \cdot 1_\{\langle W, h_{l-1}\rangle+AX_l\geq 0\} W^* \cdot h_{l-1},$ and ignore the correlation between inputs from different locations. In our method, we consider the information in $Back$ to show N-variable target functions are learnable, while [Allen-Zhu and Li, 2019a]  requires  the normalized condition to make sure  $Back\approx Back^{(0)}$ to be rougly a constant. This is one of the most different parts between this work and [Allen-Zhu and Li, 2019a].
>
>
> Meanwhile the value of $\sqrt{\widetilde{y}^T (H^\infty)^{-1}\widetilde{y}}$ is only explicitly calculated for the two-layer case in [Arora et al., 2019]. There is no existing earlier work beyond this case. In the RNN case, the neural tangenet kernel matrix involves the depth and the weight sharing in the network.  Meanwhile as mentioned in Remark 4.2, in our case, it is highly non-trivial to show $\sqrt{\widetilde{y}^T (H^\infty)^{-1}\widetilde{y}}\leq O(\mathscr{C}^*)$ with $\mathscr{C}^*$ polynomial in $L$. Our methods rely on a detailed estimation on the degeneracy of RNN with long inputs.
>
>
> **Responses to additional questions/issues:**
>
> R1:
> We apologize that the original explain of ReLU RNNs was not clear enough.
>
> The main reason why ReLU activation is rarely is that for RNNs with the gate, such as GRU or LSTM, tanh is the standard implementation. However, in this paper, we consider the simple Elman RNN with the form:
>
> $h_{l}(x)=\phi(Wh_{l-1}+AX_l).$
>
> When we use ReLU, only the hidden states $h_l$ are non-negative values. The output of RNN should be $B^Th_l$, which can be negative.
>
> For RNN with this form, it has been shown in the following works that when $\phi$ is ReLU and the initialization of $W$ is appropriate, it does work well, while tanh-RNN suffers from gradient vanishing problem:
>
> Le, Q. V. , Jaitly, N. , & Hinton, G. E. . (2015). A simple way to initialize recurrent networks of rectified linear units
> arxiv：[1504.00941]
>
> Talathi, S. S. , & Vartak, A. . (2015). Improving performance of recurrent neural network with relu nonlinearity
>
>  Li, S. , Li, W. , Cook, C. , Zhu, C. , & Gao, Y. . (2018). Independently recurrent neural network (indrnn): building a longer and deeper rnn CVPR 2018
>
>
> In fact, one of the earliest adoptions of ReLUs was on applications of RNNs for this purpose twenty years ago:
>
> Richard LT Hahnloser. On the piecewise analysis of networks of linear threshold neurons. Neural Networks, 11(4):691–697, 1998
>
> Emilio Salinas and Laurence F. Abbott. A model of multiplicative neural responses in parietal cortex. Proceedings of the National Academy of Sciences, 93(21):11956–11961, 1996.
>
> In practice, the ranking of performance is generally (under an appropriate initialization):
>
> RNN with the gate(LSTM, GRU, MGU) > ReLU-RNN >Tanh-RNN
>
> Especially, ReLU-RNN is very effective in speech recognition.
>
> R2:
>
> Thank you for your helpful suggestions!
>
> In section $H$, we show the learnability is additive. Let $y=y_1 +y_2 +...y_n$. Then
>  $\sqrt{\widetilde{y}^T H^\infty)^{-1}\widetilde{y}}\leq \sum_i \sqrt{\widetilde{y_i}^T H^\infty)^{-1}\widetilde{y_i}}$. In the two-layer case, the complexity of learning function $(\beta^T X)^k$ is proportional to $k$. Therefore we can define a complexity series. And the function is learnable if the complexity series is convergent.  Intuitionly, the complexity roughly corresponds to the norm of the target function in the reproducing kernel Hilbert space corresponding to the neural tangent kernel of the network.
>
>
> R3:
>
> We apologize that the original explain was not clear enough.
>
> We train matrix $W$ with matrices  $A$ and $B$ being  fixed during training is only for the simplicity. In our case, we care about the neural tangent kernel $H_{i,j}=\langle \nabla_A f(x_i), \nabla_A f(x_j)\rangle+ \langle \nabla_B f(x_i), \nabla_B f(x_j)\rangle+ \langle \nabla_W f(x_i), \nabla_W f(x_j)\rangle$. Note that the first two parts $\langle \nabla_A f(x_i), \nabla_A f(x_j)\rangle+ \langle \nabla_B f(x_i), \nabla_B f(x_j)\rangle$ are semidefinite. The learning ability of non-fixed $A$ and $B$ network will not be worst than the fixed one. Thus our theorem also applies to practical RNNs.
>
>  We would very much like to discuss any further questions!
>
> **References:**
>
> [Allen-Zhu et al., 2019a] Allen-Zhu, Z., Li, Y., and Liang, Y. (2019a). Learning and generalization in overparameterized neural networks, going beyond two layers
>
> [Allen-Zhu and Li, 2019a] Allen-Zhu, Z. and Li, Y. (2019a). Can sgd learn recurrent neural networks  with provable generalization?
>
> [Cao and Gu, 2019] Cao, Y. and Gu, Q. (2019). Generalization bounds of stochastic gradient descent for wide and deep neural networks.
>
> [Arora et al., 2019] Arora, S., Du, S. S., Hu, W., Li, Z., and Wang, R. (2019). Fine-grained analysis of optimization and generalization for overparameterized two-layer neural networks

---

> ### Author Response · Authors · 2021-08-18
> **We thank the reviewer for your reply!**
>
>
> We thank the reviewer for these helpful comments, while we are sorry that due to Neurips rebuttal policy, we cannot provide a revision to improve the present form during the rebuttal period.
>
> As a summary of the discussion, please allow me to emphasize that in this paper, as written in the abstract and introduction, we have already shown clearly that the main intention of this paper is to answer the following two questions:
>
> 1) How to remove the normalized condition in  [Allen-Zhu and Li, 2019a]?
>
> 2) Can RNN learn non-additive functions with the non-linear interactions between inputs in different positions?
>
> These two problems were not addressed in any previous works. And as mentioned in Remark 4.2, our results cannot be proved using the previous techniques used in  [Allen-Zhu and Li, 2019a].
>
>
>
> We apologize that the original representation was not clear enough to explain the importance of these two questions. But we believe that these two problems are no doubt substantively important for the learning ability of RNN.
>
> From the universal approximation theorem, we can show $\psi_r (\beta_r^T [X_{l_1}, X_{l_2},... X_{l_N}])$ can form a complete basis. As that in [Allen-Zhu and Li, 2019a], these results are very general.  It's natural to think trying to learn a function $f(X_1, X_2,...X_L)$ of the inputs at different positions is motivated enough. Meanwhile, in order to show the learning ability for non-additive functions, as discussed previously, our techniques are necessary.  In fact, under the normalized condition (thus the inputs have very small norms), the RNN learned in  [Allen-Zhu and Li, 2019a] is indeed a linear expansion of the inputs at $0$ rather than the RNN output we generally considered. It is because we introduced new techniques and discarded the normalized condition $||X_l||\leq \epsilon_x$ that we can prove the learning ability of RNN with inputs of the large norms and non-additive target functions. And we think the non-additive and non-linear interaction functions are what we essentially pursued from the non-linear recurrent structures.

---

### Official Review · Reviewer_Quiy · 2021-07-16

**Rating:** 6
**Confidence:** 3

**Summary:**

This paper considers when the training data are separable by certain concept class, the generalization of overparameterized RNN trained by SGD. The generalization error depends on a function complexity measure and converges at the rate $1/n$.

The treatment and results in the paper share the flavor of [Cao and Gu, 2019, Arora et al., 2019, Allen-Zhu et al., 2019]. Several comparisons are made in the paper to justify the contribution: 1) Compared to [Cao and Gu, 2019, Arora et al., 2019], no data distribution assumption is made in the current manuscript; 2) No normalized condition is needed compared to [Allen-Zhu and Li, 2019a].

Organization of the paper is clear and easy to follow, despite the proof sketch in Section 4.1 reads a bit dense. I appreciate the authors provided a proof road map in supplementary. No experiments are provided, yet the focus is purely on theory.

**Limitations And Societal Impact:**

Limitations of the work is discussed in Section 6. There is no directly foreseeable negative societal impact.

**Main Review:**

The following comments may help improve the clarity and significance of the paper:

1) A practical motivation on considering the concept class in the paper is missing. For example, does autoregressive models belong to the concept class. (I feel like they can be described by the N-variables concept class.) These examples may be helpful in explaining the data separation assumption in Theorem 2, as it is rather not a restrictive condition.

2) It is unclear how Theorem 5 and Theorem 6 are applied to deriving Equation (32). Meanwhile, Theorem 5 and 6 themselves may need some high-level explanations and intuitions on why it is true.

3) Technical novelty of the paper should be made more clear. The major contribution is establishing inequality (20). From the current presentation, it is hard to evaluate its difficulty and understand the tools used here.

---------------------- Post Author Response ----------------------------

The author response clarified most of my concerns. I would like to keep my score.

**Time Spent Reviewing:**

6 hours

---

> ### Author Response · Authors · 2021-08-06
> **Response to Reviewer Quiy**
>
> We thank the reviewer for the review and support.
>
>
> **Response to Comment 1:**
>
> We apologize that the original representation was not clear enough to explain this.
>
> In this paper, we are trying to study what kinds of functions are learnable for RNN network. We consider N-variables Concept Class with the form in (5) $\sum_r \psi_r (\beta_r^T [X_{l_1}, X_{l_2},... X_{l_N}])$. We use the form like this is because that  from Hahn-Banach theorem, or equivalent, the universal approximation theorem of neural networks with the activation function $\psi$), there exits  a suitable function $\psi$ such that $\psi(\beta^T_r [X_{l_1}, X_{l_2},... X_{l_N}])$ with different $r$ form a complete basis on the space  of all the continuous functions with input $X_{l_1}, X_{l_2}, ... X_{l_N}$. Thus $\sum_r c_r \psi(\beta^T_r [ X_{l_1}, X_{l_2}, ... X_{l_N}])$ is already a large enough kind of functions. This is why we use this form.
>
> In our N-variables Concept Class case, the target function is in the non-autoregressive form. Suppose we consider a autoregressive model:
>
> $y_{l+1}=\phi(a\cdot y_l+\beta_l^TX_l +b)$.
>
>
> When $\phi$ is a linear function, It is easy to see such a function belongs to Additive Concept Class in  (4). When $\phi(x)=\sum_i c_i x^i$, $y_{l}$ can be represented as a polynomial of $(\beta_1^TX_1, \beta_2^TX_2,... \beta_l^TX_l)$. We can show such a polynomial is learnable using the proof of learnablity of N-variables Concept Class.
>
>
>
> **Response to Comment 2:**
>
> To prove Equation (32), we need to calculate
> $\frac{1}{m}\langle \nabla_{\widetilde{W}} f(\widetilde{W},x_i),\nabla_{\widetilde{W}} f(\widetilde{W},x_j) \rangle$
>
> $=\frac{1}{m}\sum_{l,l'}\langle Back_l(x_i)\cdot D_l,Back_{l'}(x_j)\cdot D_{l'}' \rangle\cdot \langle h_l(x_i), h_{l'}(x_j)\rangle$
>
> Theorem 6 induces that if $l\neq l'$, $\frac{1}{m}\langle Back_l(x_i)\cdot D_l, Back_{l'}(x_j)\cdot D_{l'}' \rangle \approx 0$
> and
>
>  $\frac{1}{m}\langle Back_l(x_i)\cdot D_l,Back_{l}(x_j)\cdot D_{l'}' \rangle  \approx F^l_{i,j}$
>
> Meanwhile
> $F^l_{i,j}\succeq \frac{1}{K}\Sigma( \{\frac{1}{L^3}\langle X_{i,l}, X_{j,l} \rangle +K^{l-1}_{i,j}\}/Q_l)$
>
> $\langle h_l(x_i),h_l(x_j) \rangle \succeq \Omega(1) $.
>
> Proposition 2.2 says
>
> If $\phi_1(\cdot,\cdot)$ and $\phi_1(\cdot,\cdot)$ are positive definite, $\phi(x_i,x_j)=\phi_1(x_i,x_j)\cdot \phi_2(x_i,x_j)$ is also a positive definite function.
>
> Combing all these facts, directly we have $ H_{i, j} \succeq F^l_{i,j}$
>
>  $ F^l_{i,j}\succeq \frac{1}{K}\Sigma( \{\frac{1}{L^3}\langle X_{i,l}, X_{j,l} \rangle +K^{l-1}_{i,j}\}/Q_l)$.
>
> Thus (32) follows.
>
> Roughly, Theorem 6 says when we calculate
> $\frac{1}{m}\langle \nabla_{\widetilde{W}} f(\widetilde{W},x_i),\nabla_{\widetilde{W}} f(\widetilde{W},x_j) \rangle$
> $=\frac{1}{m}\sum_{l,l'}\langle Back_l(x_i)\cdot D_l,Back_{l'}(x_j)\cdot D_{l'}' \rangle\cdot \langle h_l(x_i), h_{l'}(x_j)\rangle$,
> if $l\neq l'$,
> $Back_l(x_i)\cdot D_l$ and  $Back_{l'}(x_j)\cdot D_{l'}' $ are orthogonal. Meanwhile, intuitively $Back_l$ is the decay of the information after passing through L layer. Theorem says this decay is polynomial($1/L^4$) rather than exponential.
> As mentioned in Remark 4.2, it is non-trivial.  Our methods rely on a crucial observation that the function $\lim_{l\to \infty} h_l(x_i)^Th_l(x_j)/(||h_l(x_i)||\cdot||h_l(x_j||)$ will degenerate to a constant function
>
>
>
> **Response to Comment 3:**
>
> We apologize that the original representation was not clear enough to explain the technical novelty.
>
>
>
> 1) Comparisonto [Allen-Zhu and Li, 2019a].
>
> The method in [Allen-Zhu and Li, 2019a] is to explicitly construct the approximation. Yet we use a new kernel-based viewpoint. In the supplementary material, pages 18 to 30 devote to prove Theorem 4 and only the proof of (84) is adapted from the earlier work [Allen-Zhu and Li, 2019a]. The method in this paper is more powerful so we do not use normalized conditions.
>
> 2) Comparison to [Cao and Gu, 2019, Arora et al., 2019]
>
> Similar to previous work  [Cao and Gu, 2019, Arora et al., 2019], we use an NTK based method. However, since RNNs apply the same unit repeatedly to each input token in a sequence, our analysis is significantly different from [Cao and Gu, 2019, Arora et al., 2019] case and creates lots of difficulties in the analysis.  All the 40+pages in [Allen-Zhu and Li, 2019a] devote to solve these difficulties while the two-layer case has already been well studied in [Allen-Zhu et al., 2019a].
>
>
> In fact, the value of $\sqrt{\widetilde{y}^T (H^\infty)^{-1}\widetilde{y}}$ is only explicitly calculated for the two-layer case in [Arora et al., 2019]. In the RNN case, the neural tangent kernel matrix involves the depth and the weight sharing in the network.  In  [Allen-Zhu and Li, 2019a], the method is to reduce the RNN case to $f_L\approx\sum_lBack^{(0)} \cdot 1_\{\langle W, h_{l-1}\rangle+AX_l\geq 0\} W^* \cdot h_{l-1},$
> which is similar to a summation of $L$ two-layer networks. However, this reduction requires the following complex operations in  [Allen-Zhu and Li, 2019a]:
>
> 1) Introduce new randomness to keep the independence of rows in the random initialization matrices W and A at different depths. Then estimate the perturbation.
>
> 2) Show the "off-target" Backward Correlation is zero
>
> 3) Estimate the "on target" Backward Correlation by introducing a normalized input sequence $x^{(0)}$.
>
> 4) Explicitly construct the approximation
>
>
> In this paper, we adapt 2. in [Allen-Zhu and Li, 2019a] to Eq (26). Owing to we do not use the normalized condition, we can not use any other results in  [Allen-Zhu and Li, 2019a]. To calculate the kernel matrix in this case, we use Gram-Schmidt orthonormal matrix and some Re-Randomization operations  in Theorem 5 and in section H, we introduce many new estimation.
>
> The generalizations considered in this paper, especially N-variable target functions, are very hard to be proved using the previous approximation-based methods. This is because the main idea in the previous proof in [Allen-Zhu and Li, 2019a] is to reduce the RNN function to a summation of two-layer networks as $f_L\approx\sum_l Back^{(0)} \cdot 1_\{\langle W, h_{l-1}\rangle+AX_l\geq 0\} W^* \cdot h_{l-1},$ and ignore the correlation between inputs from different locations. In our method, we consider the information in $Back$ to show N-variable target functions are learnable, while [Allen-Zhu and Li, 2019a]  requires  the normalized condition to make sure  $Back\approx Back^{(0)}$ to be rougly a constant. This is one of the most different parts between this work and [Allen-Zhu and Li, 2019a].
>
> Meanwhile, we show the polynomial decay (108) of the constant part in $Back$. As mentioned in Remark 4.2, in our case, it is highly non-trivial to show $\sqrt{\widetilde{y}^T (H^\infty)^{-1}\widetilde{y}}\leq O(\mathscr{C}^*)$ with $\mathscr{C}^*$ polynomial in $L$. Our methods rely on a detailed estimation on the degeneracy of long RNN based on Theorem 5.
>
> We would very much like to discuss any further questions!
>
>
> **References:**
>
> [Allen-Zhu et al., 2019a] Allen-Zhu, Z., Li, Y., and Liang, Y. (2019a). Learning and generalization in overparameterized neural networks, going beyond two layers
>
> [Allen-Zhu and Li, 2019a] Allen-Zhu, Z. and Li, Y. (2019a). Can sgd learn recurrent neural networks  with provable generalization?
>
> [Cao and Gu, 2019] Cao, Y. and Gu, Q. (2019). Generalization bounds of stochastic gradient descent for wide and deep neural networks.
>
> [Arora et al., 2019] Arora, S., Du, S. S., Hu, W., Li, Z., and Wang, R. (2019). Fine-grained analysis of optimization and generalization for overparameterized two-layer neural networks

---

> ### Author Response · Authors · 2021-08-23
> **Thank you for your reply!**
>
> We thank the reviewer for reading our response.  If there are any additional comments, we would very much like to hear them and improve the manuscript correspondingly.

---

### Official Review · Reviewer_kie7 · 2021-07-22

**Rating:** 6
**Confidence:** 3

**Summary:**


In this paper, the authors study the type of functions that can be learned with RNNs. They show 2 types of concept classes (additive and N-variables) that can be learned with time and sample complexity almost polynomial in L (input length). For the additive concept class, using the analysis about the NTK of the model they prove the generalization error bound for learnability of such functions without using the normalization condition. Going beyond the additive concept class, they show learnability for N-variable functions with similar guarantees.


**Limitations And Societal Impact:**

The authors discussed limitations and there is no potential negative societal impact.

**Main Review:**


The paper considers the problem of providing provable guarantees for the generalization of recurrent neural networks. The result for the additive concept class adds to the earlier work by proving the case without the normalized condition. They also go beyond the additive case and the result for N-variables functions is novel. They clearly state the differences with earlier work. I haven’t gone through every proof in the appendix but I appreciate the flow chart in the beginning. I understand that this is a theoretical paper, however, I believe adding a few numerical experiments would improve the work.

Other comments:
- Could you clarify if the result holds true for input sequences of arbitrary length?
- Line 43-46: it is not clear where the terms m and complexity are used.
- The reference links to different sections and theorems do not work.




**Time Spent Reviewing:**

5

---

> ### Author Response · Authors · 2021-08-06
> **Response to Reviewer kie7**
>
> We thank  Reviewer kie7 for your review and support!
>
>
> **Responses to Main Review:**
>
> We apologize for the faults  in Line 43-46 and the reference links.  In the revised version, we will carefully check it. In Line 43, $m$ is the width of network, and the complexity is the time and sample complexity.
>
>
> For the first question, suppose there are $n$ samples with length $l_1, l_2,...l_n\leq L_{max}$. As mentioned in Remark 3.2, our theorem aplies to "sequence labelling" task and this aply for this case. Given a label for these input sequences without padding,  RNN can learn  functions $F^*_l(x)$ for different $l$ with $F^*_l(x)$ belonging to these two types of concept classes.
>
> And the complexity is the same as that for RNN with the  $L_{max}$ length.
>
> We would very much like to discuss any further questions!

---

### Decision · Program_Chairs · 2021-09-27

**Decision:**

Accept (Poster)

**Comment:**

This paper presents new generalization bounds for RNN. While this paper follows the line of NTK theory, to deal with RNN, this paper proposes new and non-trivial techniques.